# Stretching of the retinal pigment epithelium contributes to zebrafish optic cup morphogenesis

Tania Moreno-Mármol[1,2], Mario Ledesma-Terrón[1], Noemi Tabanera[1,2], Maria Jesús Martin-Bermejo[1,2], Marcos J Cardozo[1,2], Florencia Cavodeassi[1,2†], Paola Bovolenta[1,2]*

[1]Centro de Biología Molecular Severo Ochoa, CSIC-UAM, c/ Nicolás Cabrera, 1, Campus de la Universidad Autónoma de Madrid, Madrid, Spain; [2]CIBER de Enfermedades Raras (CIBERER), Madrid, Spain

**Abstract** The vertebrate eye primordium consists of a pseudostratified neuroepithelium, the optic vesicle (OV), in which cells acquire neural retina or retinal pigment epithelium (RPE) fates. As these fates arise, the OV assumes a cup shape, influenced by mechanical forces generated within the neural retina. Whether the RPE passively adapts to retinal changes or actively contributes to OV morphogenesis remains unexplored. We generated a zebrafish Tg(E1-*bhlhe40*:GFP) line to track RPE morphogenesis and interrogate its participation in OV folding. We show that, in virtual absence of proliferation, RPE cells stretch and flatten, thereby matching the retinal curvature and promoting OV folding. Localized interference with the RPE cytoskeleton disrupts tissue stretching and OV folding. Thus, extreme RPE flattening and accelerated differentiation are efficient solutions adopted by fast-developing species to enable timely optic cup formation. This mechanism differs in amniotes, in which proliferation drives RPE expansion with a much-reduced need of cell flattening.

*For correspondence:
pbovolenta@cbm.csic.es

Present address: †St. George's, University of London, London, United Kingdom

Competing interest: The authors declare that no competing interests exist.

## Introduction

The retinal pigment epithelium (RPE) is an essential component of the vertebrate eye, composed of a monolayer of pigment-enriched epithelial cells abutting the neural retina (NR) with a primary role in photoreception (*Letelier et al., 2017*). Despite the acquisition of specialized epithelial properties, RPE cells have a neural origin and share progenitors with the NR. These progenitors are organized in a pseudostratified neuroepithelium, known as optic vesicle (OV) or eye primordium. In amniotes, the OVs appear as balloon-like structures positioned at the sides of the anterior neural tube (*Moreno-Marmol et al., 2018*). In zebrafish instead, these primordia are flat and form two bi-layered structures with the outer and inner layers distally connected by a rim or hinge (*Li et al., 2000*). Under the influence of inductive signals (*Gallardo and Bovolenta, 2018*; *Cardozo et al., 2020*), the two layers activate different genetic programs that specify the cells of the inner layer and ventral outer layer as NR and those of the dorsal outer layer as RPE (*Beccari et al., 2013*; *Buono and Martinez-Morales, 2020*; *Buono et al., 2021*). Whilst this specification occurs, the OV bends assuming a cup-like shape (*Martinez-Morales et al., 2017*).

The discovery of the *ojoplano* medaka fish mutant – affecting a transmembrane protein localized at the basal end feet of NR cells (*Martinez-Morales et al., 2009*) – in which the OV remains unfolded, was instrumental to propose that basal constriction of NR progenitors is at the basis of OV bending (*Martinez-Morales et al., 2009*). This basal constriction is mediated by the redistribution of the actomyosin cytoskeleton (*Martinez-Morales et al., 2009*; *Nicolas-Perez et al., 2016*; *Bryan et al., 2016*), which also enables the apical relaxation of retinal cells (*Sidhaye and Norden, 2017*),

**eLife digest** Rounded eyeballs help to optimize vision – but how do they acquire their distinctive shape? In animals with backbones, including humans, the eye begins to form early in development. A single layer of embryonic tissue called the optic vesicle reorganizes itself into a two-layered structure: a thin outer layer of cells, known as the retinal pigmented epithelium (RPE for short), and a thicker inner layer called the neural retina. If this process fails, the animal may be born blind or visually impaired.

How this flat two-layered structure becomes round is still being investigated. In fish, studies have shown that the inner cell layer – the neural retina – generates mechanical forces that cause the developing tissue to curve inwards to form a cup-like shape. But it was unclear whether the outer layer of cells (the RPE) also contributed to this process.

Moreno-Marmol et al. were able to investigate this question by genetically modifying zebrafish to make all new RPE cells fluoresce. Following the early development of the zebrafish eye under a microscope revealed that RPE cells flattened themselves into long thin structures that stretched to cover the entire neural retina. This change was made possible by the cell's internal skeleton reorganizing. In fact, preventing this reorganization stopped the RPE cells from flattening, and precluded the optic cup from acquiring its curved shape. The results thus confirmed a direct role for the RPE in generating curvature.

The entire process did not require the RPE to produce new cells, allowing the curved shape to emerge in just a few hours. This is a major advantage for fast-developing species such as zebrafish. In species whose embryos develop more slowly, such as mice and humans, the RPE instead grows by producing additional cells – a process that takes many days. The development of the eye thus shows how various species use different evolutionary approaches to achieve a common goal.

enhanced by focal adhesions of the apical surface with the extracellular matrix molecules (ECM) such as laminin (*Bryan et al., 2016*). The importance of concomitant apical relaxation, especially of the cells positioned at the hinge, has also been supported in studies of mammalian retinal organoids (*Eiraku et al., 2011*; *Okuda et al., 2018*). Nevertheless and independently of their relative contribution, the acquisition of apical convexity and basal concavity in the NR epithelium are accepted drivers of the biomechanical forces that induce OV folding (*Okuda et al., 2018*). In zebrafish, this mechanism is reinforced by rim involution or epithelial flow, a process whereby progenitors at the hinge emit dynamic lamellipodia at the basal side and actively translocate from the ventral outer layer of the OV into the inner/retinal layer (*Li et al., 2000*; *Sidhaye and Norden, 2017*; *Zheng et al., 2000*; *Heermann et al., 2015*; *Kwan et al., 2012*; *Picker et al., 2009*). Periocular neural crest cells appear to facilitate this flow, in part by the deposition of the ECM (*Bryan et al., 2020*) to which the lamellipodia attach (*Sidhaye and Norden, 2017*; *Heermann et al., 2015*; *Kwan et al., 2012*). The result of this flow is an unbalanced cell number between the two layers, which should favour NR bending (*Sidhaye and Norden, 2017*; *Heermann et al., 2015*; *Kwan et al., 2012*). Whether this flow may also contribute to the concomitant cell shape modifications that the remaining outer layer cells undergo as they become specified into RPE, or conversely whether RPE specification favours the flow (*Heermann et al., 2015*), remain open questions.

Indeed as the OV folds, the pseudostratified neuroepithelial cells of the OV dorsal outer layer progressively align their nuclei becoming a cuboidal monolayer in amniotes species (*Moreno-Marmol et al., 2018*; *Martinez-Morales et al., 2004*). In zebrafish, cuboidal cells further differentiate to a flat/squamous epithelium (*Zheng et al., 2000*; *Kwan et al., 2012*) that spreads to cover the whole apical surface of the NR (*Zheng et al., 2000*; *Cechmanek and McFarlane, 2017*). In mice, failure of RPE specification, as observed after genetic inactivation of key specifier genes (i.e. *Otx1/Otx2*, *Mitf*, *Yap/Taz*), enables RPE progenitors to acquire an NR fate (*Martinez-Morales et al., 2001*; *Bharti et al., 2006*; *Kim et al., 2016*). The resulting optic cups (OCs) present evident folding defects (*Martinez-Morales et al., 2001*), raising the possibility that specific RPE features are needed for OC formation. In line with this idea, a differential stiffness of the RPE vs. the NR layer has been proposed to drive the self-organization of mammalian organoids into an OC (*Eiraku et al., 2011*; *Okuda et al., 2018*; *Nakano et al., 2012*). Furthermore, generation of proper RPE cell numbers seems a requirement for

correct OC folding in mice (*Carpenter et al., 2015*). However, studies addressing the specific contribution of the RPE to OV folding are currently lacking.

Here, we report the generation of a Tg(E1-*bhlhe40*:GFP) zebrafish transgenic line with which we followed the beginning of RPE morphogenesis under both normal and interfered conditions. We show that, whereas in amniotes, including humans, the developing RPE undergo proliferation to increase its surface with a less evident cell flattening, zebrafish RPE cells rapidly cease proliferation and expand their surface by reducing their length along the apico-basal axis and extending in the medio-lateral direction with a tissue autonomous process that depends on cytoskeletal reorganization. Localized interference with either the retinal or the RPE actomyosin and microtubule cytoskeleton shows that RPE flattening generates a mechanical force that actively contributes to OV folding, complementing the force generated by the basal constriction of the NR. This mechanism represents an efficient solution to match the increased apical surface of the NR layer in a fast-developing vertebrate species such as zebrafish.

## Results

### Generation of a specific reporter line to study zebrafish RPE development

Detailed analysis of zebrafish RPE morphogenesis has been hampered by the lack of a suitable transgenic line, in which RPE cells could be followed from their initial commitment. The E40 (*bhlhe40*) gene, a basic helix-loop-helix family member, encodes a light and hypoxia-induced transcription factor (also known as *Dec1*, *Stra13*, *Sharp2*, or *Bhlhb2*) involved in cell proliferation and differentiation as well as in the control of circadian rhythms (*Yamada and Miyamoto, 2005*). In neurulating zebrafish embryos, its expression is limited to cells of the prospective RPE (*Figure 1A*; *Cechmanek and McFarlane, 2017*; *Yao et al., 2006*), representing a potentially suitable tissue marker.

We used predictive enhancer and promoter epigenetic marks at different zebrafish developmental stages (*Bogdanovic et al., 2012*) to scan the *bhlhe40* locus for the presence of conserved and active regulatory regions. The promoter and four potential enhancers (E1–4; *Figure 1B*) appeared to be active between 80 % epiboly and 24 hpf, encompassing the early stages of zebrafish eye development (*Bogdanovic et al., 2012*). These enhancers were selected, amplified, and tested using the ZED vector (*Bessa et al., 2009*) as potential drivers of gene expression in the prospective RPE. The resulting F0 embryos were raised to adulthood and screened. Only the E1 enhancer drove specific and restricted GFP reporter expression into the prospective RPE. The corresponding fishes were further crossed to establish the stable transgenic line Tg(E1-*bhlhe40*:GFP) used in this study.

Time-lapse studies of the Tg(E1-*bhlhe40*:GFP) progeny confirmed that the transgenic line faithfully recapitulated the *bhlhe40* mRNA expression profile detected with ISH (*Figure 1A and C*). GFP reporter expression appeared in a discrete group of neuroepithelial cells in the dorso-medial region of the OV (16–17 hpf) and expanded both posteriorly and ventrally (*Figure 1C*; *Figure 1—video 1* and *Figure 1—video 2*), so that, by 24 hpf, GFP-positive cells appeared to wrap around the entire inner NR layer. 3D reconstructions of selected embryos further confirmed the fast (about 7 hr) expansion of the GFP-positive domain forming an outer shell for the eye (*Figure 1D*). Apart from a faint and very transient signal in some early NR progenitors — likely due to the existence of negative regulatory elements not included in the construct — no GFP expression was observed in regions other than the RPE during this process. However, after the formation of the OC, reporter expression appeared also in the ciliary marginal zone (CMZ), the pineal gland, and few neural crest cells surrounding the eye (*Figure 1C*; *Figure 1—videos 1–3*). These additional domains of expression coincided with the reported *bhlhe40* mRNA distribution (*Yao et al., 2006*) and represented no obstacle for using the transgenic line as a tool to follow the early phases of RPE generation. Indeed, very early activation represents an important advantage of the Tg(E1-*bhlhe40*:GFP) line over other presently available transgenic lines that allow visualizing the RPE (*Zou et al., 2006*; *Miesfeld and Link, 2014*).

The suitability of the Tg(E1-*bhlhe40*:GFP) line for the identification of the very first RPE cells is supported by the onset of the reporter expression in the dorso-medial OV region, coinciding with previous fate map predictions (*Zheng et al., 2000*; *Kwan et al., 2012*). To further verify this notion, we took advantage of the characteristic of the fluorescent Kaede protein (*Ando et al., 2002*) that switches from green to red emission upon UV illumination. Embryos were injected with Kaede mRNA and neuroepithelial cells located at the most dorso-medial region of the OV were UV illuminated at

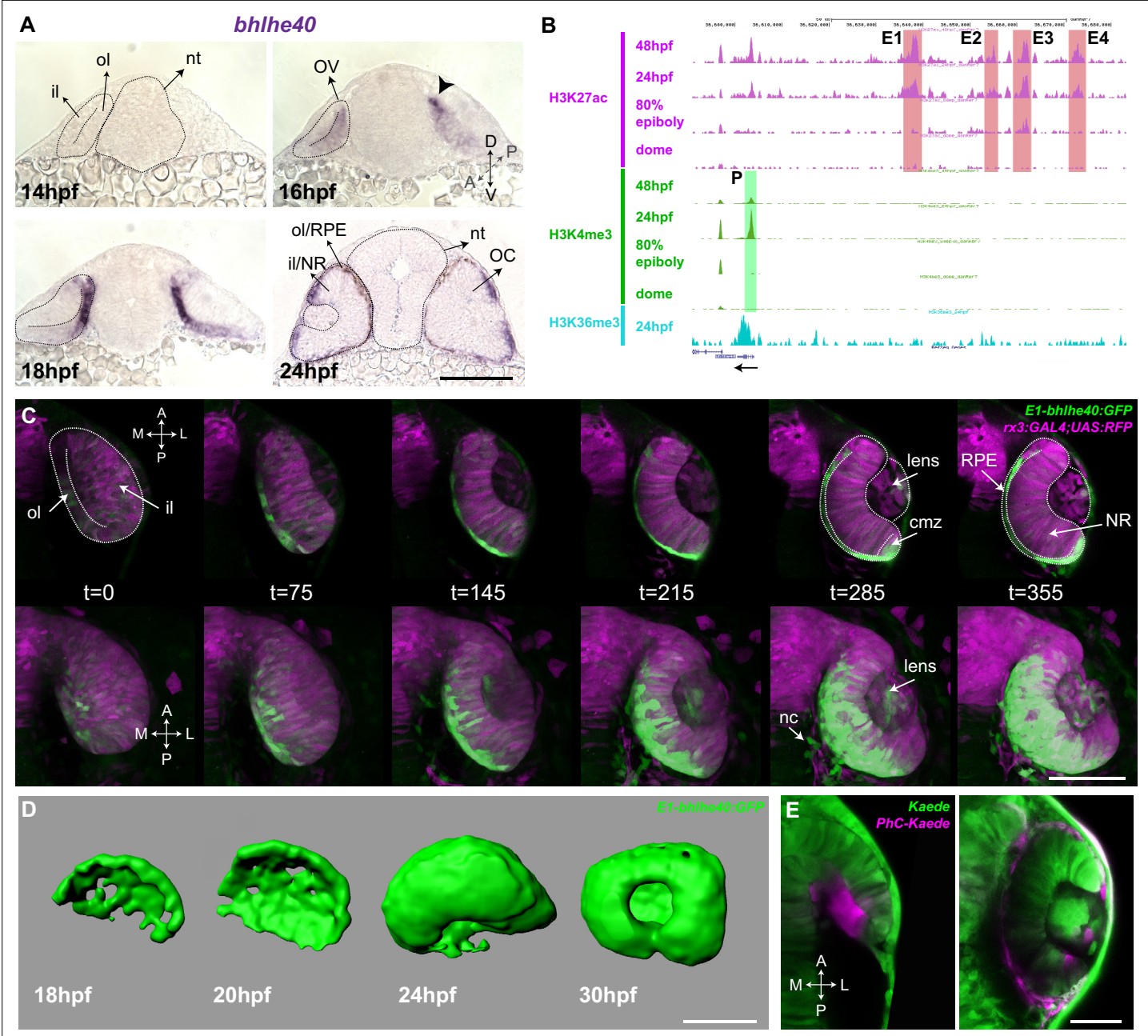

**Figure 1.** The Tg(E1-*bhlhe40*:GFP) line is a suitable tool to study early retinal pigment epithelium (RPE) generation. (**A**) Frontal cryo-sections of 14–24 hpf wild-type (wt) embryos hybridized in toto with a *bhlhe40*-specific probe. mRNA is first detected in the dorsal most region of the optic vesicle (OV) outer layer (arrowhead) and then expands ventrally. (**B**) UCSC Genome Browser view of H3K27ac (purple, potential active enhancers), H3K4me3 (green, potentially active promoters), and H3K36me3 (light blue, transcriptionally active regions) tracks obtained for four zebrafish developmental stages: dome, 80 % epiboly, 24 hpf, 48 hpf related to the upstream *bhlhe40* genomic locus (50 kb). The black arrow at the bottom indicates *bhlhe40* position and direction. The promoter (**P**) and the four selected enhancers (E1–4) are highlighted with a colour-coded box. (**C**) Time frames from in vivo time-lapse recording of a Tg(E1-*bhlhe40*:GFP;*rx3*:GAL4;UAS;RFP) embryo between 14 and 24 hpf. Time is indicated in min. Note that the GFP reporter signal matches the *bhlhe40* mRNA distribution in A. (**D**) 3D reconstruction of the prospective RPE from Tg(E1-*bhlhe40*:GFP) embryos at the stages indicated in the panel. (**E**) Dorsal view of a wt embryo injected with Kaede mRNA (green) at 12 hpf. A group of cells in the dorsal region of the outer layer was photoconverted (magenta, panel on the left) and the embryo visualized at 30 hpf (right panel). Magenta labelled cells cover the entire RPE region. Black and white dashed lines delineate the OV, neural tube, and virtual lumen in A, C. Abbreviations: A, anterior; cmz, ciliary margin zone; il, inner layer; l, lateral; m, medial; NR, NR; OC, OC; ol, outer layer; OV, optic vesicle; P, posterior; RPE, retinal pigment epithelium. Scale bars: 100 µm (**A–D**); 50 µm, E.

The online version of this article includes the following video for figure 1:

**Figure 1—video 1.** Dorsal view of the optic vesicle (OV) to optic cup (OC) transition visualized in a double Tg(E1-*bhlhe40*:GFP; *rx3*:GAL4;UAS:RFP

*Figure 1 continued on next page*

*Figure 1 continued*

embryo).

https://elifesciences.org/articles/63396/figures#fig1video1

**Figure 1—video 2.** Dorsal view of the optic vesicle (OV) to optic cup (OC) transition visualized in a double Tg(E1-*bhlhe40*:GFP; *rx3*:GAL4;UAS:RFP embryo).

https://elifesciences.org/articles/63396/figures#fig1video2

**Figure 1—video 3.** Lateral view of the optic cup (OC) folding visualized in a Tg(E1-bhlhe40:GFP) embryo injected with H2B-RFP mRNA (magenta) related to Figure 1, frame rate 1/5 min.

https://elifesciences.org/articles/63396/figures#fig1video3

the 15 hpf stage to ensure that no differentiation had yet occurred (*Figure 1E*). Embryos were let develop until 30 hpf. Photoconverted cells were found throughout the thin outer layer of the OC (*Figure 1E*), confirming that the entire RPE derives from the dorso-medial OV region.

## Neuroepithelial cell flattening drives RPE expansion at OV stages

Tg(E1-*bhlhe40*:GFP) embryos were thereafter used to dissect the extensive changes in cell shape that are associated with the acquisition of RPE identity (*Zheng et al., 2000*; *Cechmanek and McFarlane, 2017*). At OV stage all retinal progenitors present a columnar-like morphology characteristic of embryonic neuroepithelia (*Figure 2A and A'*). As soon as RPE progenitors begin to express the transgenic GFP reporter, their apico-basal length rapidly and progressively reduces (*Figure 2A–C'*), so that the cells first assume a cuboidal shape (*Figure 2B and B'*) and then become flat, forming a squamous epithelial monolayer overlaying the apical surface of the NR (*Figure 2C and C'*). At 30 hpf, RPE cells presented a polygonal, frequently hexagonal, morphology (*Figure 2D and D'*), with an apical surface area that, on average, became about eightfold larger than that observed in progenitor (PN) cells (*Figure 2F*; RPE⁻a: $354.8 \pm 100.3$ µm² vs. PN⁻a: $43.7 \pm 7.8$ µm²). In contrast, the abutting apical surface of NR cells slightly shrank as compared to that of PN cells (*Figure 2E, E' and F*; NR⁻a: $22.5 \pm 2.9$ µm² vs. PN⁻a: $43.7 \pm 7.8$ µm²) while maintaining a constant apico-basal length. The latter observation agrees with previous reports showing that the cone-like morphology of NR progenitors represents only a slight modification of the progenitor columnar shape (*Nicolas-Perez et al., 2016*; *Sidhaye and Norden, 2017*).

To obtain a quantitative analysis of the dynamic changes that RPE tissue, as whole, underwent during OV folding, we performed a morphometric characterization of the images from *Figure 1—videos 1–3*. To this end, the fluorescent information from the Tg(E1-*bhlhe40*:GFP) reporter was discretized into seven different segments that were individually analysed along the recording time (*Figure 3—figure supplement 1*; Materials and methods). The combined quantification of the different segments (*Figure 3A*; *Figure 3—figure supplement 1*) showed that, between stages 17 and 21 hpf, the overall thickness of the RPE tissue underwent, on average, a flattening of more than threefold (from a mean of about 24–8 µm; *Figure 3B*). Flattening occurred with a central to peripheral direction, so that RPE cells closer to the hinges were the last ones to flatten (*Figure 2C and C'*). In parallel, the overall RPE surface underwent an approximately two fold expansion between 17 and 22 hpf (from approximately $1.1–2.2 \times 10^3$ µm²; *Figure 3C*; *Figure 1—video 1* and *Figure 1—video 2*), reflecting the large increase in the apical area observed in each individual cell at later stages (*Figure 2*). In line with the idea that cell flattening is per se sufficient to account for whole tissue enlargement, the RPE volume only slightly changed between 17 and 20 hpf with a slope increase of $0.47 \times 10^3$ µm³/h (*Figure 3D*).

To provide further support to this idea, we analysed the RPE volume variation in comparison with the growth of the entire OC in two time windows: from 17 to 22 hpf (*Figure 1—videos 1–3*) and from 24 to 37 hpf (*Figure 3—video 1*), using GFP (RPE) and RFP (eye) reporter signals from the double Tg(E1-*bhlhe40*:GFP; *rx3*:GAL4;UAS;RFP) line or from the Tg(E1-*bhlhe40*:GFP) line injected with the pCS2:*H2B-RFP* mRNA (*Figure 3E and F*). Signal quantification showed that the eye underwent a marked and linear volume increase (slope: $5.54 \times 10^4$ µm³/hr from 17 to 22 hpf and $3.6 \times 10^4$ µm³/hr from 24 to 37 hpf) as compared to that of the RPE (*Figure 3D and G*). Between 20 and 22 hpf the reporter starts being expressed in the posterior and, to a lesser extent, in the anterior CMZ (GFP-CMZ domain, *Figure 1—videos 1 and 2*). Consistently with the onset of GFP-CMZ expression, RPE reporter volume suddenly expanded between 20 and 22 hpf (slope: $1.25 \times 10^4$ µm³/hr; *Figure 3G*) to then slow back between 24 and 37 hpf (slope: $1.2 \times 10^3$ µm³/hr; *Figure 3G*). Confirming this association, only the

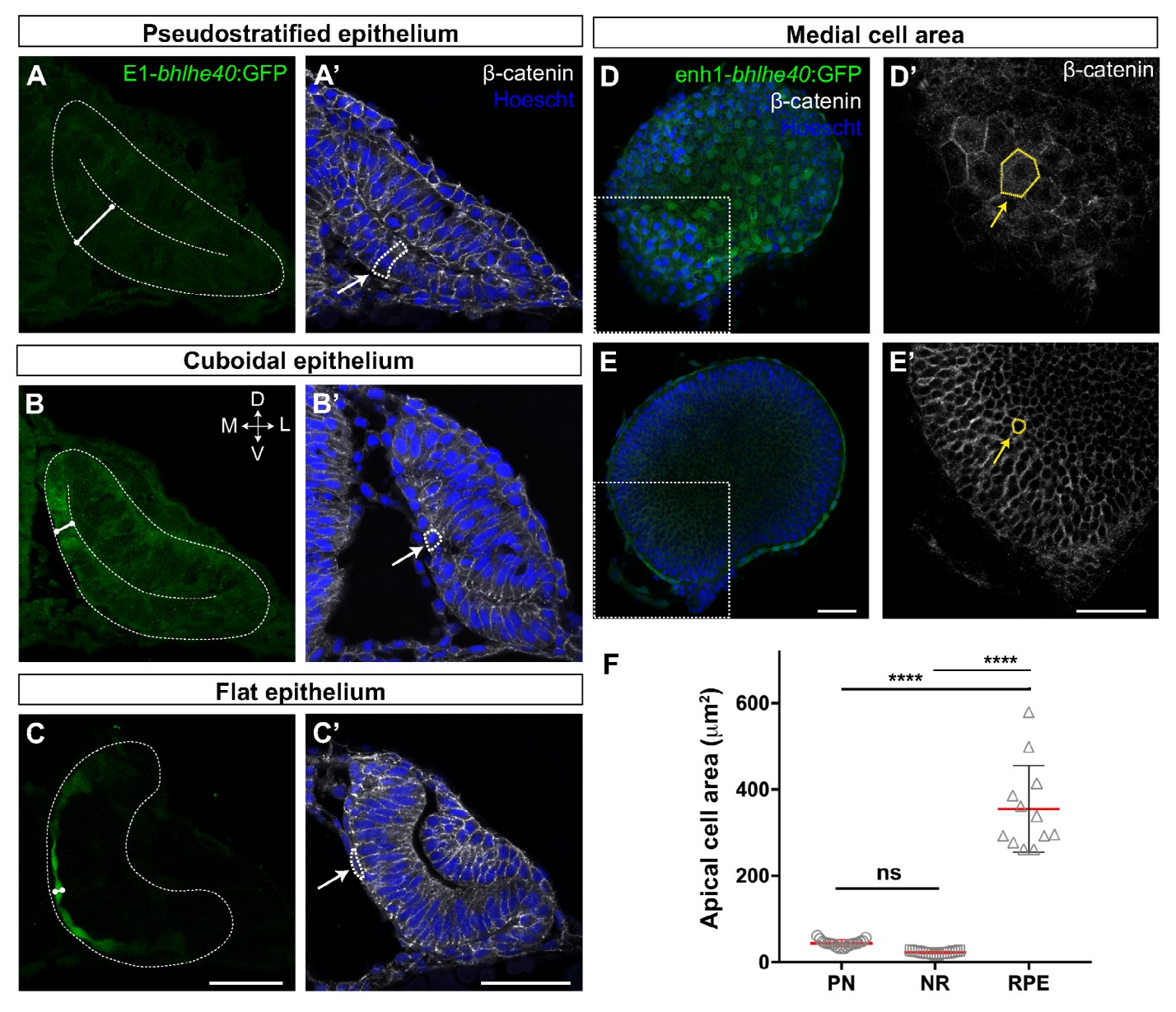

**Figure 2.** The retinal pigment epithelium (RPE) converts from a pseudostratified to a squamous epithelium during optic vesicle (OV) folding by increasing individual cell surface. (**A–C'**) Confocal images of frontal cryo-sections of Tg(E1-*bhlhe40*:GFP) embryos immunostained for GFP (green) and β-catenin (white) and counterstained with Hoechst (blue). Note that the RPE rapidly decreases its thickness white straight line in (**A–C**) and cells change from columnar (14 hpf, arrow in A') to cuboidal (16 hpf, arrow in B') and then flat shape (22 hpf, arrow in C'). White dashed lines delineate eye contour and virtual lumen in A–C. (**D–E'**) Confocal images of the posterior RPE (**D, D'**) and neural retina (NR) (**E, E'**) regions of an eye cup dissected from 30 hpf Tg(E1-*bhlhe40*:GFP) embryos immunostained for GFP (green) and β-catenin (white) and counterstained with Hoechst (blue). Images in D', E' are high power views of the areas boxed in white box in D, E. Note the hexagonal morphology (yellow arrow in D') of RPE cells (average area 354.8 ± 100.3 µm²) in contrast to the small and roundish cross-section of retinal progenitors (average area 22.5 ± 2.9 µm²; yellow arrow in E'). (**F**) The graph represents the average area of individual OV progenitors and NR and RPE cells (n = 15–19). The average area is calculated using cells from five different embryos. Data represent mean ± SD, ****p < 0.0001. ns, non-significant. Scale bar: 50 µm.

The online version of this article includes the following source data for figure 2:

**Source data 1.** The source data 1 is the excell file that is already correcly linked.

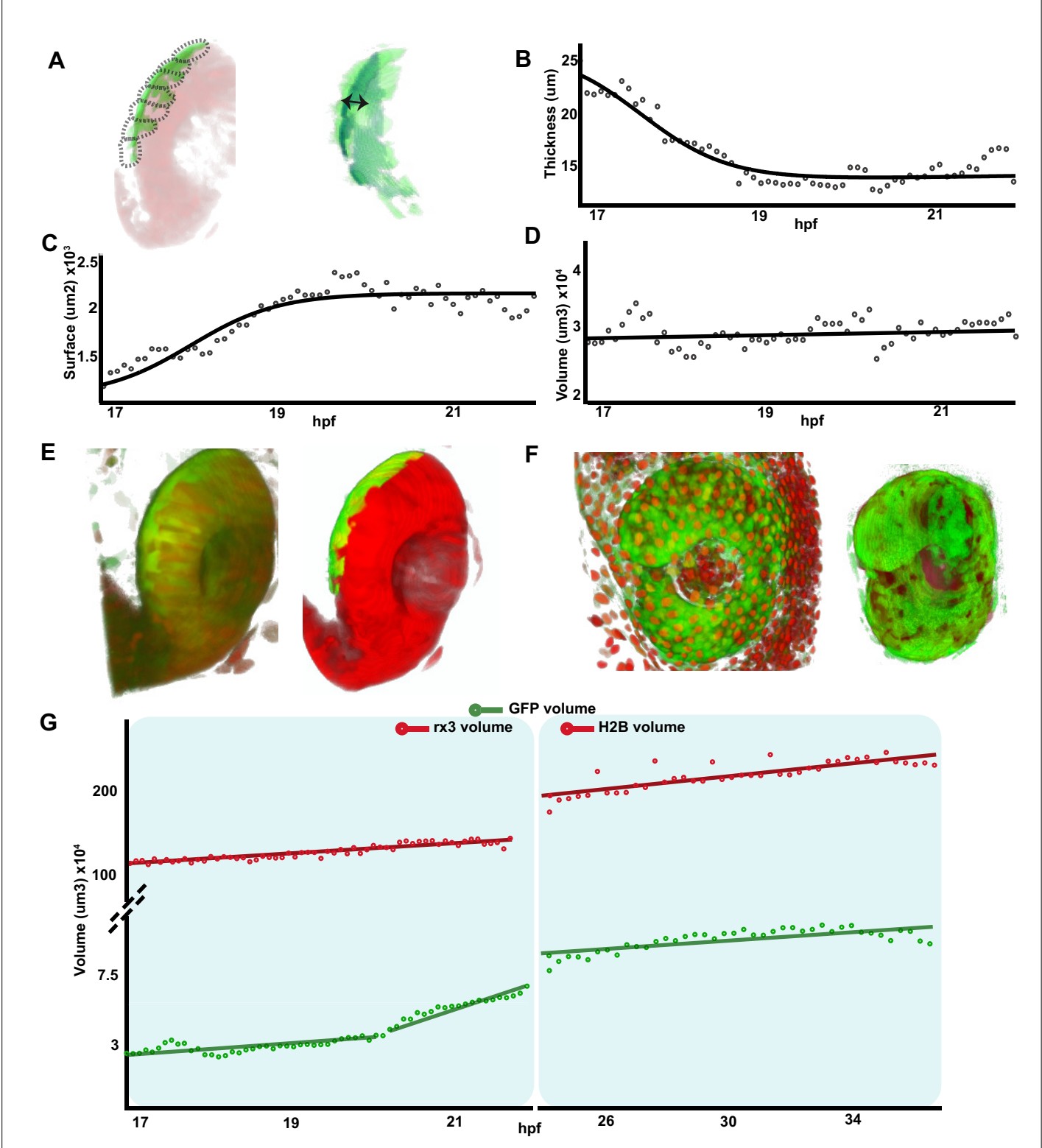

**Figure 3.** Retinal pigment epithelium (RPE) volume is conserved during initial tissue morphogenesis. (**A**) Image on the left represents the reconstruction of a single frame from *Figure 1—video 2* Tg(E1-*bhlhe40*:GFP; *rx3*:GAL4;UAS:RFP embryo) showing the optic vesicle/optic cup (OV/OC) in red and the RPE in green. The segments in which the RPE was discretized are depicted with black dashed lines. The image on the right shows the RPE reconstruction obtained after filtering. Double arrow points to RPE thickness. (**B–D**) The graphs show how the RPE thickness (B, calculated as volume/surface), surface (**C**), and volume (**D**) change as a function of the developmental stage. (**E**) 3D reconstructions of raw (left) and processed (right) versions

*Figure 3 continued on next page*

*Figure 3 continued*

of a frame from *Figure 1—videos 1 and 2*. (**F**) 3D reconstructions of raw (left) and processed (right) versions of a frame from *Figure 3—video 1*. (**G**) Quantification of RPE and eye volume based on *Figure 1—videos 1 and 2* (rx3 volume quantification) and *Figure 3—video 1* (H2B volume quantification) along developmental stages.

The online version of this article includes the following video, source data, and figure supplement(s) for figure 3:

**Source data 1.** Quantification of RPE thickness along time and space.

**Figure supplement 1.** Retinal pigment epithelium (RPE) region selection from the GFP-positive domain.

**Figure supplement 1—source data 1.** Quantification of RPE volume in space and time.

**Figure 3—video 1.** Lateral view of optic cup (OC) growth visualized in a Tg(E1-*bhlhe40*:GFP) embryo injected with H2B-RFP mRNA (magenta), related to Figure 3, frame rate 1/5 min.

https://elifesciences.org/articles/63396/figures#fig3video1

tissue segments very close to the posterior CMZ had a volume larger than that of the RPE at 17–20 hpf (*Figure 3—figure supplement 1*), whereas the GFP-positive RPE domain located in the most central regions presented a volume undistinguishable from that detected at previous stages. In sum, a comparison of the dynamics slopes from the GFP-RPE domain and OV regions suggests that the volume of the RPE grows at very low pace ($0.47 \times 10^3$ μm³/hr) – despite the rather drastic morphological changes of its cells – whereas the whole OV expands at a pace ~ 25 times faster ($1.25 \times 10^4$ μm³/hr; *Figure 3D*).

Taken all together, this morphometric analysis indicates that the expansion of the RPE in zebrafish occurs by recruiting a limited number of cells that undergo profound cell shape changes: from a neuroepithelial to squamous morphology.

## RPE flattening is a tissue autonomous process required for proper OV folding

Both external interactions and intracellular processes determine the shape of a cell and define its mechanical properties (*Totaro et al., 2018*). Thus, in principle, RPE flattening might occur as a 'passive' process, triggered by the forces that the NR and hinge cells exert on the RPE (*Moreno-Marmol et al., 2018*; *Heermann et al., 2015*). Alternatively, it might depend on cell or tissue autonomous cytoskeletal rearrangements, involving, for example, myosin II activity, which controls the acquisition of a flat epithelial morphology in other contexts (*Tee et al., 2011*; *Vishavkarma et al., 2014*). Discriminating between these two possibilities has been technically difficult. Experiments directed to assess the mechanisms of OV folding have used whole embryo bathing in drugs such as blebbistatin (*Nicolas-Perez et al., 2016*; *Sidhaye and Norden, 2017*), a specific myosin II inhibitor (*Rauscher et al., 2018*). Such an approach hampers the assessment of the potential influence of NR over RPE morphogenesis (and vice versa) as well as the relative contribution of the two tissues to OV folding. We sought to overcome this limitation by spatially localized interference with the cytoskeletal organization of either the RPE or NR and by recording the tissue autonomous and non-autonomous consequences. Nevertheless, to begin with, we reproduced the whole embryo bathing approach used by others (*Nicolas-Perez et al., 2016*; *Sidhaye and Norden, 2017*), focusing on the yet unreported effect that blebbistatin had on the RPE.

Tg(E1-*bhlhe40*:GFP) embryos were bathed either in blebbistatin or its diluent (DMSO) at 17 hpf (the onset of RPE specification; *Figure 4A–C*) and then let develop up to 19.5 hpf, when embryos were analysed. DMSO-treated (control) embryos developed normally forming an OC surrounded by a squamous RPE (*Figure 4B*). In blebbistatin-treated embryos, NR cells did not undergo basal constriction and the OV remained unfolded (*Figure 4C*), as previously described (*Nicolas-Perez et al., 2016*; *Sidhaye and Norden, 2017*). Notably, in almost all the embryos analysed (n = 44/49), RPE cells did not flatten but remained cuboidal in shape (*Figure 4C*). A similar phenotype was observed after treatment with paranitroblebbistatin, a non-cytotoxic and photostable version of blebbistatin (*Figure 4D*). These observations support that lack of OV folding is associated with alterations in both the retina and RPE. To uncouple the two events, we turned to the photoactivable compound azidoblebbistatin (Ableb), which binds covalently to myosin II upon two-photon irradiation, thus permanently interfering with myosin II activity in a spatially restricted manner, as already proven (*Kepiro et al., 2012*; *Kepiro et al., 2015*). Tg(E1-*bhlhe40*:GFP) 17 hpf embryos were bathed in Ableb or in DMSO and irradiated in a

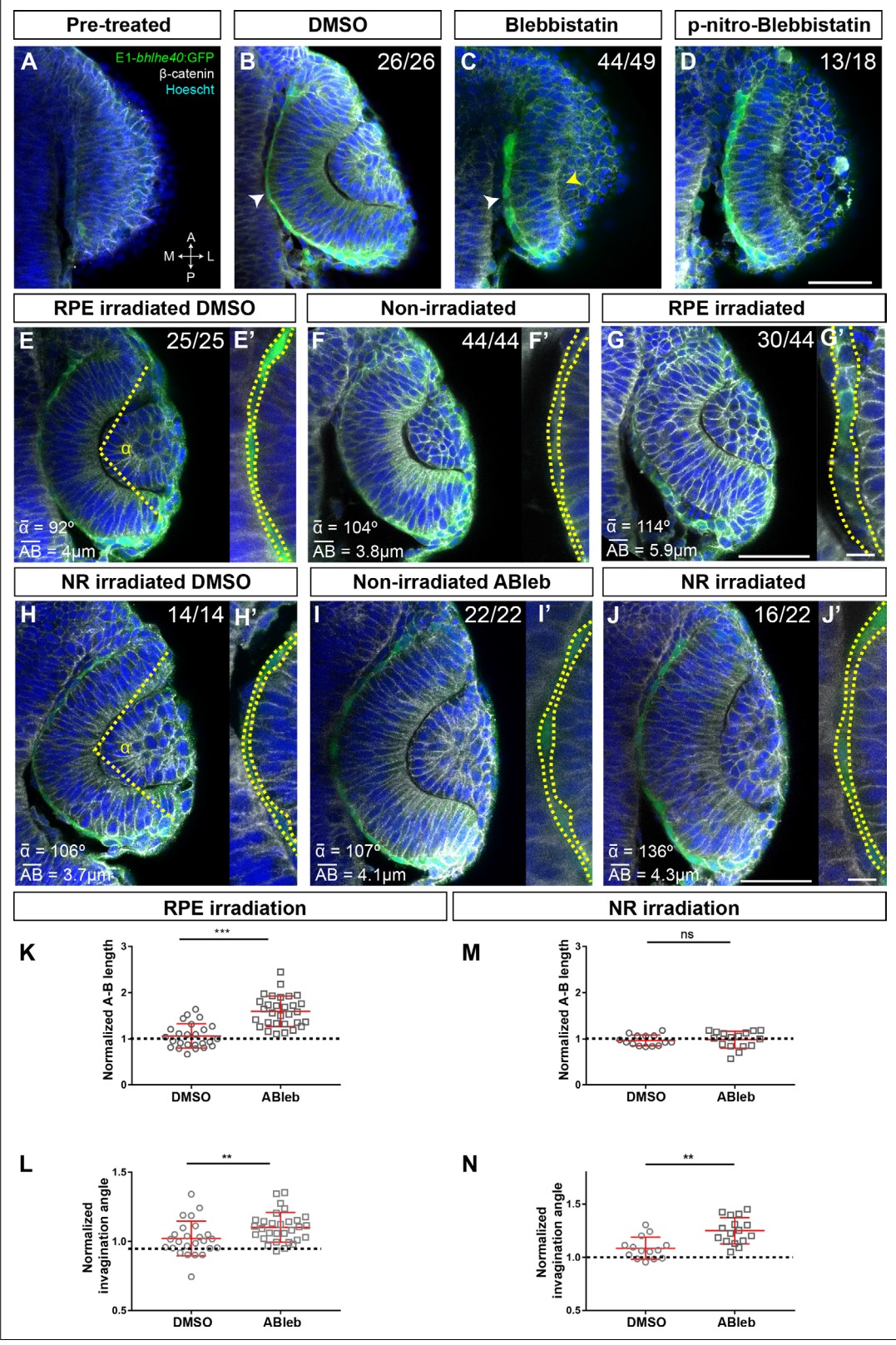

**Figure 4.** Retinal pigment epithelium (RPE) flattening is a myosin-dependent cell autonomous process required for proper optic vesicle (OV) folding. (**A–J**) Confocal images of dorsally viewed Tg(E1-*bhlhe40*:GFP) embryos before (17 hpf; **A**) and 2.5 hr after incubation (19.5 hpf) with either DMSO (**B, E, H**), blebbistatin (**C**), paranitroblebbistatin (**D**), or azidoblebbistatin (Ableb) (**F, G, I, J**) with (**G, J**) or without irradiation (**F, I**) in the prospective RPE (**F–G**) or

*Figure 4 continued on next page*

*Figure 4 continued*

neural retina (NR) (**I–J**). Images in E', F', G' H', I', and J' are high power views of RPE morphology. Embryos were immunostained for GFP (green), β-catenin (white), and counterstained with Hoechst (blue). Note that the optic cup (OC) forms and the RPE flattens (white arrowhead in B) normally in all DMSO-treated embryos (B, E, E,' H, **H'**) or in embryos incubated in Ableb without irradiation (**F, F', I, I'**). In contrast, the RPE remains cuboidal (white arrowhead in C) and NR cells seem not undergo basal constriction (yellow arrowhead in C) in the presence of myosin inhibitors (**C, D**). Photoactivation of Ableb in the RPE prevents cell flattening (compare E', F', with G') and impairs OV folding (**G**). When Ableb is photoactivated in the NR, folding of the OV is also impaired (**J**) but RPE cells undergo flattening (compare H', I', with J'). The number of embryos analysed and showing the illustrated phenotype is indicated on the top right corner of each panel and the average invagination angle and mean A–B on the left bottom corner. The yellow dashed line in (**E, H**) indicates how the invagination angle (α) was determined. (**K, M**) Normalized RPE height in DMSO- and Ableb-treated embryos, irradiated either in the RPE (**K**) or in the NR (**M**). (**L, N**) Normalized invagination angle in DMSO- and Ableb-treated embryos irradiated either in the RPE (**L**) or in the NR (**N**). Data represent mean ± SD; **p < 0.01 and ***p < 0.001. ns, non-significant. Scale bars: 50 µm in A–J and 25 µm in E'–J'.

The online version of this article includes the following source data for figure 4:

**Source data 1.** Quantification of A-P length and invagination angles reported in *Figure 4K–N*.

small region of either the dorsal outer layer (RPE) or the inner retinal layer (retina) of the OV (see Materials and methods). Embryos were then let develop until 24 hpf. During this period, the irradiated RPE cells underwent anterior and medio-lateral spreading – likely coinciding with the reported pinwheel 'movement' (*Kwan et al., 2012*) – and were mostly found in the ventral half of the OV. Notably, Ableb photoactivation in the prospective RPE cells reproduced, although slightly less efficiently (n = 30/44 embryos), the phenotype observed upon whole embryo bathing in blebbistatin, in which RPE cells acquired a cuboidal morphology (*Figure 4C, D, G, G'*). No detectable alterations were found in the OV of irradiated/DMSO-treated embryos or in the contralateral non-irradiated OV of embryos incubated in Ableb regardless of the irradiated region (*Figure 4E, F', H, I'*). Cell shape quantifications showed a significantly longer apico-basal axis (*Figure 4E'–G'*) in irradiated Ableb RPE cells, normalized to that of control (DMSO- and Ableb-treated non-irradiated) OVs (*Figure 4K*; Mann-Whitney U test, z = −5.088, p < 0.001, control mean length 15.96 vs. Ableb-treated 38.03). Failure of cell flattening in the irradiated region of the RPE was consistently associated with a significant reduction of OV folding (*Figure 4E–G*), as assessed by measuring the invagination angle (*Sidhaye and Norden, 2017*), which was normalized to that of control embryos (*Figure 4L*; Mann-Whitney U test: z = −2.704, p < 0.01, mean rank for control 21.60 vs. Ableb-treated 33.33). Photoactivation of Ableb in similar areas of the prospective NR basal region resulted in an elongated NR and a significantly impaired OV folding (*Figure 4J*), as determined by the invagination angles normalized to those of control OV

(*Figure 4N*; Mann-Whitney U test: z = −3.035, p < 0.01, mean rank control 10.29 vs. Ableb 20.06). Notably, disruption of NR morphogenesis had no consequences on RPE development in all the analysed embryos (n = 16/22): cells underwent normal flattening with apico-basal lengths comparable to those of controls (*Figure 4H'–J' and M*; Mann-Whitney U test: z = 0.582, p > 0.05, mean rank control 14.50 vs. Ableb 16.38). These data strongly support that RPE flattening is not secondary to NR folding but rather a tissue autonomous event. They also indicate that OV folding requires forces independently generated in both the NR and RPE. Notably, blebbistatin or Ableb treatments did not compromise the expression of the Tg(E1-*bhlhe40*:GFP) transgene in any experimental condition, indicating that cellular tension and morphology did not affect RPE specification.

Microtubule dynamics has an important role in determining the shape of a cell (*Yevick and*

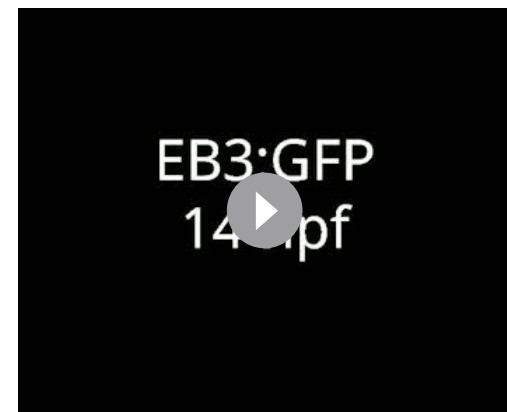

**Video 1.** *eb3*GFP dynamics 14, 17, and 23 hpf, when retinal pigment epithelium (RPE) cells have a neuroepithelial (continuous acquisition, n = 9), cuboidal (continuous acquisition, n = 10), or squamous conformation (continuous acquisition, n = 9).
https://elifesciences.org/articles/63396/figures#video1

*Martin, 2018*). For example, reorientation of the microtubule cytoskeleton from the apico-basal to the medio-lateral cell axis together with actin filaments redistribution seems to drive the conversion of the *Drosophila* amnioserosa cells from a columnar to squamous epithelium (*Pope and Harris, 2008*). To determine if a similar reorientation occurs in the RPE, we used time-lapse analysis of Tg(E1-*bhlhe40*:GFP) embryos injected with the mRNA of EB3:GFP, a protein that binds to the plus end of growing microtubules (*Stepanova et al., 2003*). In neuroepithelial RPE progenitors, microtubules grew in the apico-basal direction, whereas growth turned to the medio-lateral plane, as the RPE cells became squamous (*Figure 5—figure supplement 1*; *Video 1*). To determine if this reorientation is important for cell flattening, we bathed Tg(E1-*bhlhe40*:GFP) embryos in nocodazole, a drug that interferes with microtubule polymerization, or its vehicle (DMSO) at either 16 or 17 hpf (*Figure 5A and D*) and then analysed them at 18.5 or 19.5 hpf, respectively. The eye of DMSO-treated embryos developed normally (*Figure 5B and E*), whereas in the presence of nocodazole RPE cells retained a columnar-like morphology with a stronger phenotype in embryos exposed to the drug at an earlier stage (*Figure 5C*). Nocodazole treatment did not prevent the activation of the GFP reporter expression (*Figure 5C*) or the acquisition/distribution of expected specification (*otx1* and *mitf*) and apico-basal polarity (zo-1 and laminin) markers (*Figure 5—figure supplement 2*). Notably, although the NR layer appeared to bend inward, the RPE layer remained unfolded (*Figure 5F*) and outer layer cells accumulated at the hinge, suggesting a defect in rim involution. This defect may be due to the alteration of microtubule polymerization in rim cells. Alternatively, the lack of RPE stretching may prevent the translocation of rim cells to the NR layer.

The whole embryo treatments described above did not allow us to determine the differential requirement of microtubule dynamics in the RPE and the adjacent NR layer. However, we were unable to uncouple the effect of microtubule alterations in the two OV layers with localized drug interference. We thus resorted to use stathmin 1 (*STMN1*), a key regulator of microtubule depolymerization (*Belmont and Mitchison, 1996*). We generated a bidirectional UAS construct (UAS:*STMN1*) driving the simultaneous production of GFP and *STMN1* under the same regulatory sequences (*Paquet et al., 2009*; *Distel et al., 2010*), which we injected in Tg(*rx3*:GAL4) embryos. We reasoned that, although *rx3* drives transgene expression in both NR and RPE progenitors, the random and sparse expression that occurs in F0 would be sufficient to separate the effect in the two tissues. RPE cells expressing *STMN1* – and notably also those nearby – retained a cuboidal-like shape with an abnormally increased apico-basal axis as compared to GFP-positive cells in control UAS:GFP-injected embryos. Even though cells in the inner OV layer appeared to still undergo basal constriction (*Figure 5G–I*), the OV as a whole underwent poor invagination (*Figure 5J*).

All in all, the data derived from the manipulation of the actomyosin and microtubule cytoskeleton suggest that the RPE actively participates in OV folding by undergoing a tissue autonomous stretching driven by cell cytoskeletal rearrangements.

## Differential requirement of cell proliferation in zebrafish vs. amniotes RPE development

Our finding that the zebrafish RPE largely grows through autonomous cell flattening agrees with the observation that zebrafish RPE cells barely proliferate during OV folding (*Cechmanek and McFarlane, 2017*). Furthermore, pharmacological treatment of embryos to block cell division during OV folding has little or no consequences on RPE expansion (*Cechmanek and McFarlane, 2017*). These observations however differ from reports in mouse embryos, in which RPE proliferation seems a requirement for OV folding (*Carpenter et al., 2015*) and suggest the existence of species-specific modes of early RPE growth. We hypothesized that these modes may be related to the speed of embryonic development with final consequences on the epithelial characteristic of the RPE. To test this possibility, we compared proliferation rate and apico-basal length of the zebrafish RPE (*Figure 6*) with those of the medaka, chick, mouse, and human embryos at equivalent OV/OC stages (*Figure 7*). In these species, Otx2 and N-cadherin immunostaining was used to identify the RPE domain (*Martinez-Morales et al., 2001*; *Bovolenta et al., 1997*) and the cell shape (*Figure 7—figure supplement 1*), respectively.

5-Bromo-2′-deoxyuridine (BrdU) incorporation in Tg(E1-*bhlhe40*:GFP) embryos from early OV (17 hpf) to late OC stages (48 hpf) showed a marked reduction of cell proliferation in the OV outer layer (*Figure 6A–B*), very much in line with the report that only 2 % of the outer layer cells undergo mitosis during this period (*Cechmanek and McFarlane, 2017*). At the earlier stage (17 hpf), BrdU-positive

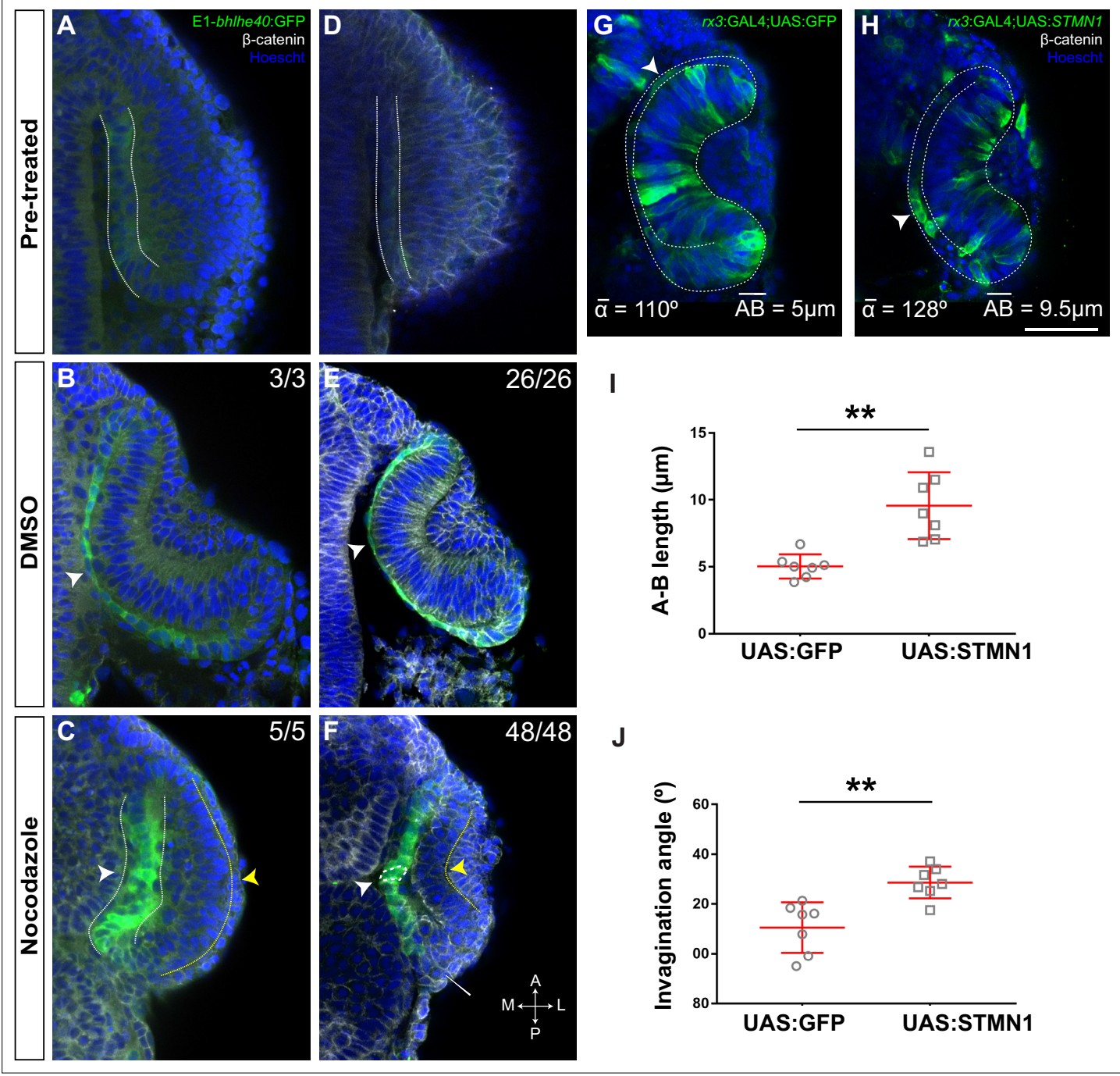

**Figure 5.** Microtubule dynamics is required for retinal pigment epithelium (RPE) cell flattening and optic vesicle (OV) folding. (**A–F**) Confocal images of dorsally viewed Tg(E1-*bhlhe40*:GFP) embryos before (16 hpf A; 17 hpf D) and 2.5 hr after incubation (18.5 hpf B, C; 19.5 hpf E, F) with either DMSO (**B, E**) or nocodazolef (**C, F**). Embryos were immunostained for GFP (green, **A–F**), β-catenin (white, **D–F**) and counterstained with Hoechst (blue). Note that the optic cup (OC) forms and the RPE flattens (white arrowhead in B, **E**) normally in DMSO-treated embryos. RPE cells retain a columnar-like morphology in the presence of nocodazole (white arrowhead in C, **F**). In embryos treated at earlier stage, the neural retina (NR) seems to bend outward (yellow arrowhead in C), whereas some folding occurs when the embryos are treated at later stages (yellow arrowhead in F), although cells seem to accumulate at the hinge (thin white arrowhead, **F**). The number of embryos analysed and showing the illustrated phenotype is indicated on the top right corner of each panel. (**G, H**) Confocal images of dorsally viewed *rx3*:GAL4;UAS;RFP embryos injected with the UAS:GFP (G, n = 7) or UAS:*STMN1* (H, n = 7) at one cell stage and fixed at 24 hpf. Embryos were labelled with anti-GFP (green) and counterstained with Hoechst (blue). Note that *STMN1* overexpression but not GFP prevents RPE cell flattening and cells retain a cuboidal-like shape (white arrowheads in G–H). The average invagination angle and mean length of the A–B axis are indicated in the bottom left and right angles, respectively. (**I, J**) The graphs show the length of the A–B axis (**I**) and the invagination angle (**J**) in embryos overexpressing GFP or *STMN1*. Mean ± SD. \*\*p < 0.01. Scale bar: 50 µm.

*Figure 5 continued on next page*

*Figure 5 continued*

The online version of this article includes the following source data and figure supplement(s) for figure 5:

**Source data 1.** Quantification of A-P length and invagination angle of the experimentes reported in *Figure 5I,J*.

**Figure supplement 1.** EB3-GFP dynamics during retinal pigment epithelium (RPE) cell remodelling.

**Figure supplement 2.** Nocodazole treatment does not alter retinal pigment epithelium (RPE) specification and polarity.

cells were scattered across the RPE with no easily identifiable geometry and accounted for 49 % of the

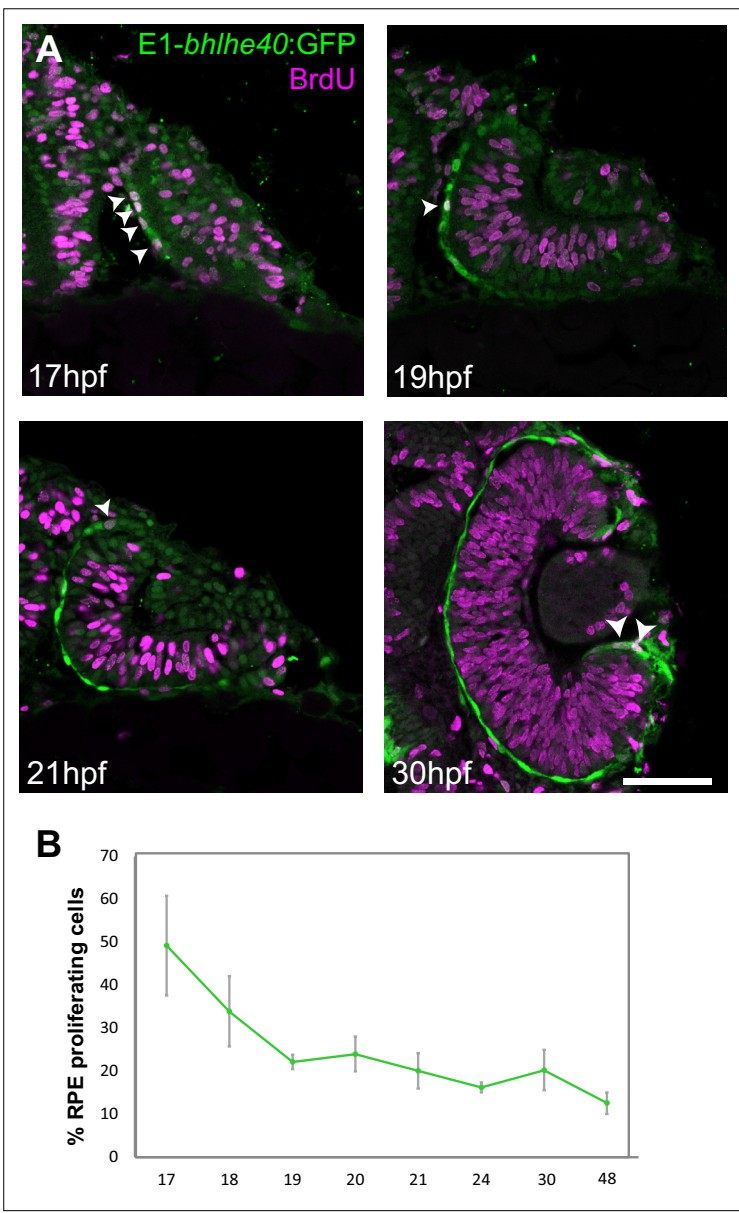

**Figure 6.** Zebrafish retinal pigment epithelium (RPE) flattening is associated with an abrupt decrease of cell proliferation. (**A**) Confocal images of dorsally viewed Tg(E1-*bhlhe40*:GFP) embryos exposed to 5-bromo-2'-deoxyuridine (BrdU) at different developmental stages as indicated in the panel and immunostained for BrdU (magenta) and GFP (green). (**B**) Percentage of RPE proliferating cells (BrdU+/total Hoechst +) in 17–48 hpf Tg(E1-*bhlhe40*:GFP) embryos. Mean ± SD; n = 5 embryos per stage. Scale bar: 100 µm.

The online version of this article includes the following source data for figure 6:

**Source data 1.** Quantification of the RPE proliferating cells at diffrent developmental stages as reported in *Figure 6B*.

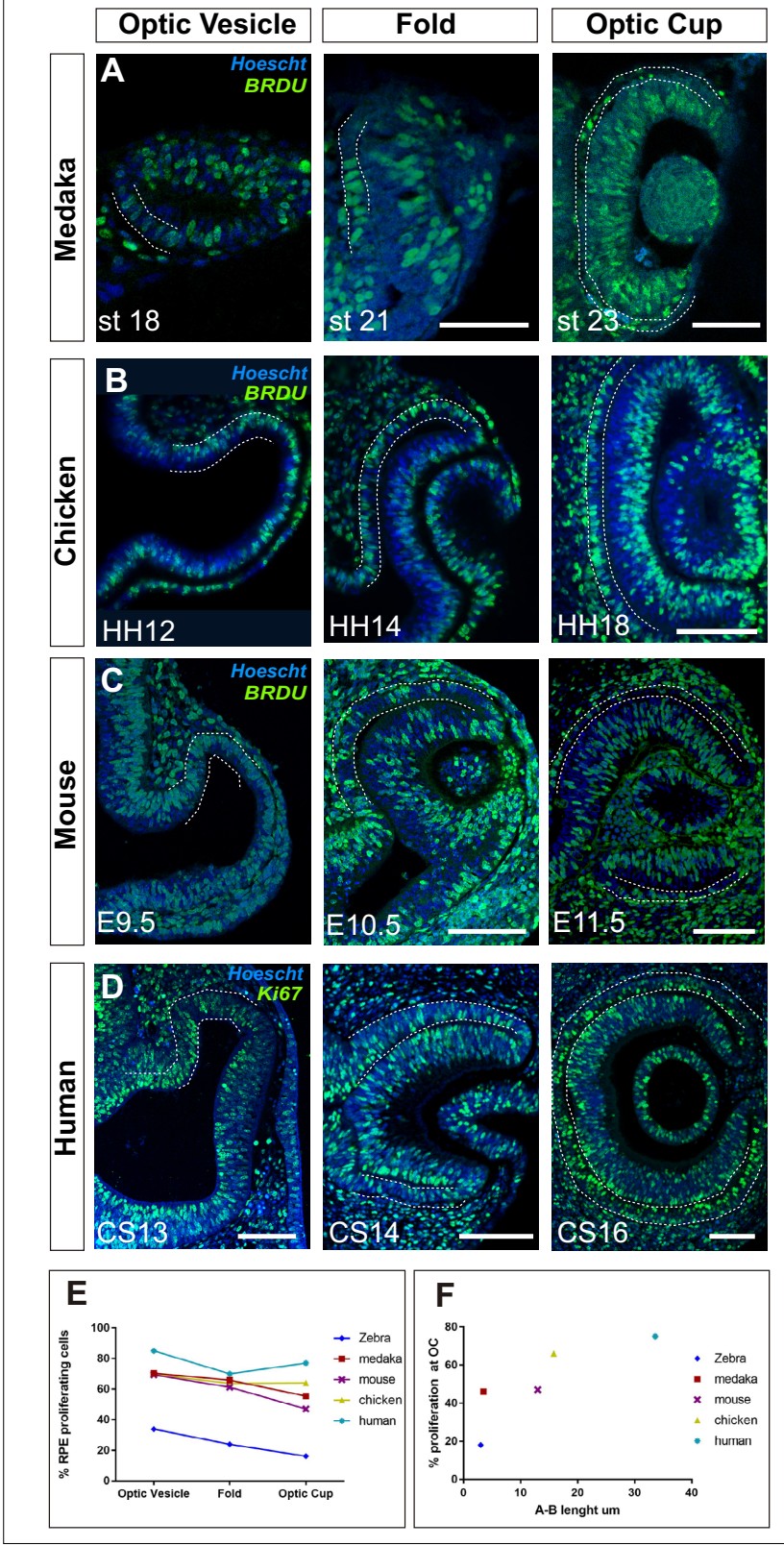

**Figure 7.** Proliferation accounts for retinal pigment epithelium (RPE) surface increase during amniotes optic vesicle (OV) folding. (**A–C**) Confocal images of frontal sections from medaka, chick, and mouse embryos exposed to 5-bromo-2'-deoxyuridine (BrdU) at equivalent stages of OV folding into optic cup (OC), as indicated in the panels. Sections were immunostained for BrdU (green) and counterstained with Hoeschst (blue). In all panels, the

*Figure 7 continued on next page*

*Figure 7 continued*

prospective RPE has been highlighted with dotted white lines on the basis of the Otx2 immunostaining illustrated in *Figure 7—figure supplement 1*. (**D**) Confocal images of horizontal sections from human embryos at equivalent stages of OV folding into OC. Sections were immunostained for Ki67 (green) and counterstained with Hoeschst (blue). (**E**) Percentage of RPE proliferating cells (BrdU+/total Hoechst+) in the analysed period and compared to those reported in *Figure 6B* for zebrafish. Data represent mean ± SD; the number of embryos analysed for each stage varied between 3 and 10. (**F**) Relationship between proliferation rate and apico-basal axis length at OC stage in the different species. Note that there is a positive correlation between the two parameters. Scale bars: 50 μm in A and 100 μm in B–D.

The online version of this article includes the following source data and figure supplement(s) for figure 7:

**Source data 1.** Data supporting graphs in *Figure 7E,F*.

**Figure supplement 1.** Identification of the RPE domain and RPE cell shape in amniotes.

total RPE cells. This fraction dropped to about 20 % at 19 hpf, when cells are flat, and then to 12 % at 48 hpf (*Figure 6B*) when the epithelium is maturing. Statistical analysis showed significant differences between 17 and 20 hpf (Mann-Whitney U test: z = −2.619, p < 0.01, mean rank for 17 hpf is 8 and for 19 hpf is 3) and a clear correlation between proliferation rate and developmental stage (Kruskal-Wallis test: $\chi^2$(df = 7, n = 40) = 32,023; p < 0.001). During this period, the apico-basal axis of individual RPE cells flattened reaching a length of 3 μm at 22–23 hpf. Thus, acquisition of RPE identity, cell shape changes, and OV folding are associated with a progressive reduction of cell proliferation in the OV outer layer.

OC morphogenesis in the teleost medaka fish occurs with a choreography comparable to that of the zebrafish (*Heermann et al., 2015*; *Kwan et al., 2012*) but the medaka fish RPE does not adopt an extreme squamous morphology (*Figure 7A*). Notably, medaka fish develop slower than zebrafish embryos, so that, from first appearance, their OVs take about 8 hr more to reach a fully developed OC (26 vs. 18 hr) (*Furutani-Seiki and Wittbrodt, 2004*), a time compatible with an additional round of cell division. Consistent with this idea, BrdU incorporation in st18 to 22–23 medaka embryos showed that about 70 % of the cells in the OV outer layer were actively cycling and this proportion dropped to about 48 % at OC stage (*Figure 7A and E*) with a slightly less evident decrease of the average apico-basal axis (st18: 21.3 μm vs. st23 3.5 μm; *Figure 7A*) as compared to the changes observed in zebrafish. An equivalent analysis in chick and mouse embryos showed similar results. In these species OV conversion into an OC takes about 27 and 48 hr, respectively. During this period a similar and almost constant proportion of RPE progenitors incorporated BrdU (*Figure 7B, C and E*), including when cells acquired the expression of the RPE differentiation marker Otx2 (*Figure 7—figure supplement 1*). Furthermore, RPE cells only roughly halved their apico-basal axis (chick: HH12: 30.1 μm vs. HH18: 15.8 μm; mouse: E9.5: 23.7 μm vs. E11.5 13 μm *Figure 7B and C*), suggesting that in slower developing species, proliferation but not stretching accounts for RPE surface increase. To corroborate this idea, we next analysed human embryos.

The human eye primordium is first visible at about 4–5 weeks of gestation corresponding to Carnegie stage (CS)13 (*O'Rahilly, 1983*). A fully formed OC is reached only roughly 10 days after, at CS16 (*O'Rahilly, 1983*). Immunostaining of paraffin sections from CS13 to CS16 embryos with antibodies against Ki67, a marker of the active phases of the cell cycle, demonstrated that the large majority of prospective RPE cells undergo a marked proliferation during the transition from OV to OC (*Figure 7D*). Owing to the difficulties in obtaining early human embryonic samples, the percentage of proliferating cells could only be estimated, showing that in the OTX2-positive domain (*Figure 4*, *Figure 7—figure supplement 1B*), Ki67-positive RPE cells represented about 85–75% of the total between CS13 and CS16. During this period, the prospective RPE layer always appeared as a rather thick pseudostratified epithelium with an organization resembling that of the NR composed of densely packed and elongated neuroepithelial cells (*Figure 7D*; *Figure 7—figure supplement 1*). During the formation of the OC, the RPE neuroepithelium only slightly flattened (apico-basal thickness: CS13: 45 μm vs. CS16: 33.6 μm), far from reaching the cuboidal appearance seen at postnatal ages (*Figure 7—figure supplement 1*).

Collectively, these data indicate that, in the absence of sufficient time for cell proliferation, flattening is an efficient solution adopted by zebrafish RPE cells to enlarge the whole tissue to the extent needed for OV folding. In other vertebrates, in which slower development allows for more rounds

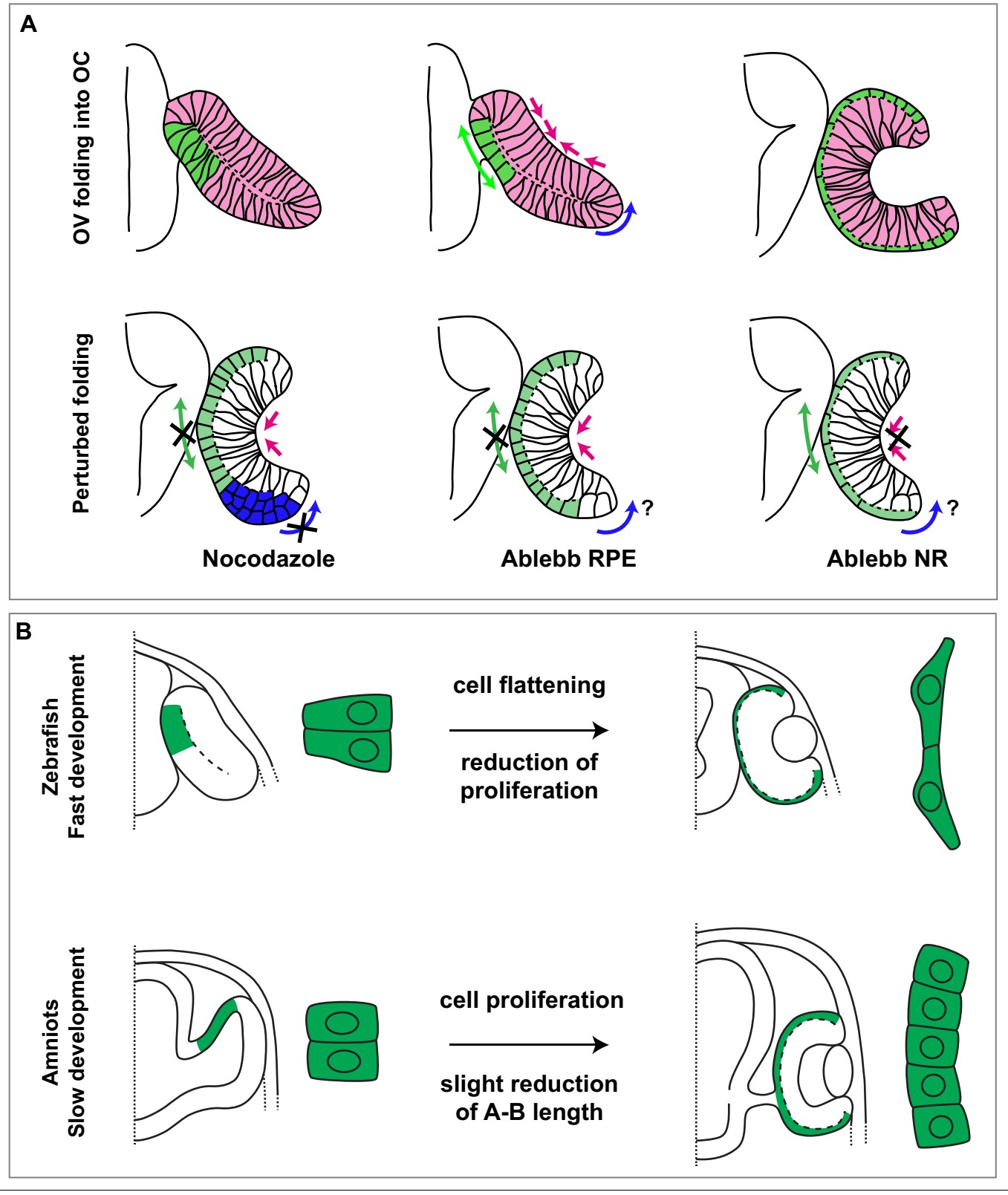

**Figure 8.** Summary of species-specific modes of retinal pigment epithelium (RPE) differentiation and its contribution to optic vesicle (OV) folding. (**A**) The drawing on the top represent the dynamic of OV folding into an optic cup (OC). Green double arrow indicated RPE flattening, blue arrow rim involution whereas pink arrows indicate retinal basal constriction. Bottom row summarizes the alterations in OV folding observed after localized interference with RPE and neural retina (NR) cytoskeleton. (**B**) Schematic representation of the differential mechanisms by which the RPE in zebrafish

*Figure 8 continued on next page*

Figure 8 continued

(upper row) and in amniotes (lower row) expands its surface during OV folding morphogenesis. In zebrafish, the RPE enlarges its surface by cell stretching; in amniotes, including in humans, the RPE instead expands by cell proliferation with a less pronounced need of cell flattening.

of cell division, the RPE grows in a conventional proliferation-based mode that correlates with a less evident flattening of RPE cells (*Figure 7F*).

## Discussion

The cup shape of the vertebrate eye is thought to optimize vision (*Goldsmith, 1990*). This shape is acquired very early in development as the result of specification and morphogenetic events, during which the NR and the RPE arise. Studies in teleosts (zebrafish and medaka) together with mammalian organoid cultures have recently demonstrated a fundamental contribution of NR progenitors in driving the acquisition of this cup shape (*Moreno-Marmol et al., 2018*; *Martinez-Morales et al., 2017*). The role of the RPE progenitors in this process has instead not been properly clarified. In this study, we have filled this gap and analysed the folding of the zebrafish OV from the RPE perspective. This analysis has been possible thanks to the generation of a new RPE reporter line Tg(E1-*bhlhe40*:GFP), in which GFP expression appears in the domain fated to originate the RPE. Following the cells arising from this domain, we show that RPE surface expansion is an active and tissue autonomous process required for OV folding. This expansion largely occurs by extreme cell flattening with little contribution of cell proliferation, a mechanism that sets zebrafish RPE morphogenesis apart from that of other analysed vertebrate species, in which proliferation accounts for RPE growth.

Our analysis together with a previous report (*Cechmanek and McFarlane, 2017*) shows that the onset *bhlhe40* expression coincides spatially and temporally with that of zebrafish RPE specification. Thus, the Tg(E1-*bhlhe40*:GFP) line serves as an early tissue-specific marker that even precedes the appearance of previously accepted *Otx* or *Mitf* tissue specifiers, as confirmed in a parallel transcriptomic analysis (*Buono et al., 2021*). *Bhlhe40* expression in the RPE is conserved at least in mouse and humans (*Buono et al., 2021*; *Cohen-Tayar et al., 2018*; *Hu et al., 2019*), suggesting a possible relevant function in this tissue. However, its CRISP/Cas9 inactivation, alone or in conjunction with that of the related *bhlhe41, mitfa,* and *mitfb*, had no evident consequences on zebrafish RPE development, at least in our hands (data not shown). One possible reason for the absence of an evident RPE phenotype is functional redundancy with other untested members of the large family of the BHLH transcription factors or that the gene has only later functions as reported (*Abe et al., 2006*). However, we favour the alternative possibility that zebrafish RPE specification does not occur stepwise as in other species (*Martinez-Morales et al., 2004*; *Fuhrmann et al., 2014*) but 'en bloc' with an almost simultaneous activation of all differentiation genes. This would make the inactivation of one or two genes insufficient to perturb fate acquisition. Such a mechanism is expected to provide robustness to a process that takes place in just few hours and finds support in present and past findings (*Buono et al., 2021*; *Cechmanek and McFarlane, 2017*).

Indeed, we and others *Cechmanek and McFarlane, 2017* have shown that, by the time the OV starts to bend, the large majority of RPE cells have already left the cell cycle and have acquired a differentiated squamous morphology by undergoing a marked surface enlargement in the medio-lateral direction and a reduction of the apico-basal axis. The net result is an overall modest volume increase. Furthermore, blocking cell division as the OC forms does not interfere with RPE expansion (*Cechmanek and McFarlane, 2017*), strongly supporting a primary role of cell stretching in RPE expansion. Consistently, transcriptomic analysis shows that during this same lag of time, RPE cells repress genes characteristic of 16 hpf OV progenitors, such as *vsx1*, and acquire the expression of RPE-specific genes. These include blocks of transcription factors, such as known RPE specifiers (i.e. *otx, mitf*) and regulators of epidermal specification (i.e. *tfap* family members, known regulator of keratin gene expression; *Leask et al., 1991*) as well as several cytoskeletal components, most prominently a large number of keratins and other desmosomal components found in squamous epithelia (*Buono et al., 2021*). Thus, in just few hours (from 16 to 18 hpf) RPE cells acquire the molecular machinery required for their conversion from a neuroepithelial to a squamous and likely highly coupled epithelium. Our study shows that this conversion relays on a tissue autonomous cytoskeletal reorganization without the influence of the morphogenetic events occurring in the nearby NR. Indeed, local interference with actomyosin or microtubule dynamics is sufficient to retain RPE cells into a cuboidal

or neuroepithelial configuration, respectively, without affecting their specification. In contrast, localized interference with NR bending has no effect on RPE flattening. Notably, our studies also suggest that the RPE acts in a 'syncytial-like' manner, as mosaic interference with microtubule polymerization seems to impact in the shape of the adjacent cells if not on the entire tissue. This is perhaps not surprising given that mature RPE cells have been reported to be chemically coupled (*Pearson et al., 2004*; *Bao et al., 2019*). Furthermore, the presumptive RPE of the chick (unpublished observations) and zebrafish (*Buono et al., 2021*) expresses high levels of connexion proteins (i.e. Gap-43), which are responsible for the 'syncytial-like' behaviour observed in brain astrocytes (*Buskila et al., 2019*). This together with the additional observation that st18 RPE cells express many desmosomal proteins (*Buono et al., 2021*) indicate that the tissue becomes tightly connected very soon, perhaps behaving as a community (*Gurdon, 1988*).

The extreme flattening of the zebrafish RPE cells makes the resolution of their cytoskeletal components difficult with in vivo confocal microscopy, hampering the complete understanding of how the actomyosin cytoskeleton promotes the acquisition of a squamous configuration. In other contexts, a flat morphology is associated with the presence of acto-myosin stress fibres that compress the nucleus (*Tee et al., 2011*; *Vishavkarma et al., 2014*). Myosin II is essential for this compressive role and its inhibition with blebbistatin causes the loss of the flat morphology (*Tee et al., 2011*; *Vishavkarma et al., 2014*), as we have observed in blebbistatin- and Ableb-treated embryos. It is thus possible that a similar nuclear compression may occur in the RPE cells as they flatten, although we were unable to detect stress fibres around the nucleus, likely due to plasma membrane proximity. Remodelling of the microtubular cytoskeleton seems to aid further RPE cell flattening. Microtubules change their orientation during RPE morphogenesis, from being aligned along the apico-basal axis of the cells at the onset of RPE morphogenesis, to becoming aligned with the planar axis in squamous RPE cells. A similar process has been described during the morphogenesis of the *Drosophila* amnioserosa (*Pope and Harris, 2008*), in which cells also change from a columnar to a squamous morphology. In these cells, actin accumulation at the apical edge seems to provide resistance to the elongation of microtubules, which thus bend, leading to a 90° rotation of all subcellular components. This rotation is accompanied by a myosin-dependent remodelling of the adherens junctions (*Pope and Harris, 2008*), a process that may also take place during RPE flattening.

Although additional studies are needed to clarify the precise dynamics of the cytoskeletal reorganization underlying RPE differentiation, our study demonstrates that cytoskeletal dynamics occurs in a tissue autonomous manner. In contrast to other studies (*Nicolas-Perez et al., 2016*; *Sidhaye and Norden, 2017*), we have used a photoactivable version of blebbistatin that has allowed us to determine the individual contribution of the NR and RPE to OV folding. As a drawback, this approach allows to activate the drug only in relatively small patches of tissue. It was thus rather remarkable to observe that failure of RPE flattening in small regions was sufficient to decrease OV folding. This suggests that RPE stretching represents an additional and relevant mechanical force that, together with retinal basal constriction and rim involution, contributes to zebrafish eye morphogenesis (*Figure 8A*). This flattening and stretching together with a substantial expression of keratins (*Buono et al., 2021*) may confer a particular mechanical strength to the zebrafish RPE, which, in turn, may constrain the NR at the same time favouring rim involution (*Heermann et al., 2015*). The latter possibility is supported by the observation that inner layer cells seem to accumulate at the hinge in the absence of RPE flattening. Alternatively, this accumulation may simply reflect that rim cell involution depends on intrinsic microtubule polymerization, although previous studies have discarded this possibility (*Sidhaye and Norden, 2017*). These marked morphogenetic rearrangements can thus be seen as an efficient solution adopted in fast-developing species to make eye morphogenesis feasible in a period that does not allow for proliferation-based tissue growth.

The perhaps obvious question is whether similar morphogenetic rearrangements are needed in other vertebrates to form the remarkably conserved cup shape of the eye. So far, rim involution has been reported only in teleost species where it may represent a fast mode of increasing the surface of the inner layer of the OV, thus favouring its bending (*Sidhaye and Norden, 2017*; *Heermann et al., 2015*; *Kwan et al., 2012*). This idea is well in agreement with previous data showing that between 16 and 27 hpf the number of cells in the outer layer of the OV decreases from about 587 to 432, whereas that of the inner layer increases in a way that cannot be explained solely by proliferation (*Zheng et al., 2000*). In other species, this cell displacement may not be needed as the layer can

grow by cell division. In a similar way, we have shown here that in slower developing species, RPE cells maintain a higher proliferation rate that contributes substantially to the increase of RPE surface while undergoing less marked changes in cell shape (*Figure 8B*). This correlation is visible in medaka, despite its relative evolutionary proximity to zebrafish (*Furutani-Seiki and Wittbrodt, 2004*), and is maximal in human embryos. Indeed, in humans, the RPE layer is composed of cells with a neuroepithelial appearance and a high proliferation rate, despite the expression of OTX2, considered a tissue specifier. Thus, in mammals, full commitment of the OV outer layer to an RPE identity may occur over a prolonged period of time and not 'en bloc' as in zebrafish, as suggested by comparing RNA-seq data of RPE cells from human CS13–16 embryos (*Hu et al., 2019*) with those from equivalent stages in zebrafish (*Buono et al., 2021*). Human RPE cells from CS13 to CS16 embryos are still enriched in the expression of proliferation associated genes (*Hu et al., 2019*) but not of those typical of squamous epithelia as in zebrafish (*Buono et al., 2021*). A slow acquisition of RPE identity may also explain why, in mice, inactivation of genes such as *Otx2, Mitf,* or *Yap* causes the RPE layer to adopt NR characteristic (*Martinez-Morales et al., 2001*; *Kim et al., 2016*; *Nguyen and Arnheiter, 2000*), whereas this feature that has never been reported after equivalent manipulations in zebrafish (*Lane and Lister, 2012*; *Miesfeld et al., 2015*), or why FGF8 can push the amniote but not the zebrafish RPE layer to acquire an NR identity (*Martinez-Morales et al., 2005*). As a reflection of this slower differentiation in amniotes, RPE cells can largely retain their neuroepithelial morphology and adopt a final cuboidal – but not squamous – appearance at a slower and species-specific pace.

We thus propose that RPE cell stretching vs. cell addition are different solutions adopted by species with different rates of development to reach a common goal: an appropriate equilibrium between the surface of the RPE and that of the NR. Indeed, the present study together with previous observations (*Carpenter et al., 2015*) and in silico models (*Okuda et al., 2018*; *Eiraku et al., 2012*) support that this equilibrium is a prerequisite for proper OV folding.

## Materials and methods

**Key resources table**

| Reagent type (species) or resource | Designation | Source or reference | Identifiers | Additional information |
|---|---|---|---|---|
| Gene (*Danio rerio*) | bhlhe40 | ENSEMBL | ENSDARG0000004060 | Ref. 28 |
| Strain, strain background (*Oryzias latipes*) | Wild-type, adult cab strain | CBMSO fish room | | NBRP Medaka (https://shigen.nig.ac.jp/medaka/) |
| Strain, strain background (*Mus musculus*) | Wild-type BALB/c | CBMSO animal facility | | https://www.jax.org/jax-mice-and-services |
| Strain, strain background (*Danio rerio*) | Adult wild-type AB/tupl strain | CBMSO fish room | | ZIRC (https://zebrafish.org/home/guide.ph) |
| Genetic reagent (*Danio rerio*) | Tg(E1-bhlhe40:GFP) | Transgenic line generated in this study | | Details in Materials and methods, 'Generation of the Tg(E1-bhlhe40:GFP) line' section |
| Genetic reagent (*Danio rerio*) | Tg(rx3:Gal4-VP16;UAS:GFP) | PMID:22819672 | ZFIN Cat# ZDB-GENO-121105-83, RRID:ZFIN_ZDB-GENO-121105-83 | Ref. 48 |
| Biological sample (*Homo sapiens*) | Paraffin sections of human embryonic eye primordia | Human Dev. Biology Resource (http://www.hdbr.org/) | | |
| Recombinant DNA reagent | ZED vector | PMID:19653328 | | Ref. 31 |
| Recombinant DNA reagent | Bidirectional UAS:GFP | PMID:19363289 | | Ref. 45 |
| Recombinant DNA reagent | pQTEV-STMN1 | Addgene# 31326 | RRID:Addgene_31326 | |

*Continued*

| Reagent type (species) or resource | Designation | Source or reference | Identifiers | Additional information |
|---|---|---|---|---|
| Recombinant DNA reagent | UAS: STMN1 | Construct generated in this study | | Details in Materials and methods, 'Gal4-UAS-mediated expression' |
| Recombinant DNA reagent | pCS2-Kaede | PMID:17406330 | | Ref. 34 |
| Recombinant DNA reagent | pCS2-H2b-mRFP | Addgene# 53745 | RRID:Addgene_53745 | |
| Recombinant DNA reagent | pCS2-EB3-GFP | PMID:12684451 | | Ref. 43 |
| Antibody | Anti-BrdU (mouse) | Becton-Dickinson | | IF(1:200), |
| Antibody | Anti-GFP (chicken polyclonal) | Abcam | Cat# ab13970, RRID:AB_300798 | IF(1:2000) |
| Antibody | Anti-βcatenin (mouse monoclonal) | BD Transduction Laboratories | Cat# 610153, RRID:AB_397554 | IF(1:400) |
| Antibody | Anti-ZO1 (rabbit monoclonal) | Invitrogen | | IF(1:400) |
| Antibody | Anti-laminin (rabbit polyclonal) | Sigma | Cat# L9393, RRID:AB_477163 | IF(1:200) |
| Antibody | Anti-otx2 (rabbit polyclonal) | Abcam | Cat# ab76748, RRID:AB_1524130 | IF(1:1000) |
| Antibody | Anti-Ki67 (rabbit polyclonal) | Abcam | Cat# ab15580, RRID:AB_443209 | IF(1:500) |
| Commercial assay or kit | GatewayTM LR ClonaseTM Enzyme Mix | Invitrogen | 11791019 | |
| Commercial assay or kit | pCR8/GW/TOPO TA Cloning Kit | Invitrogen | K250020 | |
| Commercial assay or kit | mMessage mMachine SP6 transcription kit | Invitrogen | AM1340 | |
| Commercial assay or kit | NucleoSpin RNA Clean-up kit | Macherey Nagel | 740948.50 | |
| Chemical compound, drug | Blebbistatin | Calbiochem | Blebbistatin-CAS674289-55-5-Calbioche, | 100 µM |
| Chemical compound, drug | Paranitroblebbistatin | Optopharma | DR-N-111 | 20 µM |
| Chemical compound, drug | Azidoblebbistatin | Optopharma | DR-A-081 | 5 µM |
| Chemical compound, drug | Nocodazole | Sigma | M1404 | 10 ng/µl |
| Chemical compound, drug | BrdU | Roche | B23151 | 5 mg/ml |
| Software, algorithm | SPSS | CSIC bioinformatic resources | RRID:SCR_002865 | IBM (https://www.ibm.com/uk-en/products/spss-statistics) |
| Software, algorithm | MATLAB | CSIC bioinformatic resources | RRID:SCR_001622 | MathWorks (https://www.mathworks.com/products/get-matlab.htm) |
| Other | DAPI stain | Invitrogen | D1306 | |

## Animals

Adult zebrafish (*Danio rerio*) were maintained under standard conditions at 28 °C on 14/10 hr light/dark cycles. AB/Tübingen strain was used to generate the transgenic lines and as control wild type. Embryos and larvae were kept in E3 medium (5 mM NaCl, 0.17 mM KCl, 0.33 mM CaCl$_2$, 0.33 mM

MgSO$_4$) supplemented with Methylene Blue (Sigma) at 28 °C and staged according to somite number and morphology (*Kimmel et al., 1995*). The Tg(E1-*bhlhe40*:GFP) and Tg(*rx3*:Gal4;UAS:RFP) (*Weiss et al., 2012*) lines were maintained in the same conditions and crossed to generate the Tg(E1-*bhlhe40*:GFP;*rx3*:GAL4;UAS;RFP) line. Wild-type medaka fish (*Oryzias latipes*) of the cab strain were maintained at 28 °C on a 14/10 hr light/dark cycle. Embryos were staged as described (*Iwamatsu, 2004*). Fertilized chick embryos (Santa Isabel Farm, Cordoba, Spain) were incubated at 38 °C in a humidified rotating incubator until the desired stage. Embryos were inspected for normal development and staged according to *Hamburger and Hamilton, 1992*. Wild-type BALB/c mice were in pathogen-free conditions at the CBMSO animal facilities, following current national and European guidelines (Directive 2010/63/EU). The day of the appearance of the vaginal plug was considered as embryonic day (E)0.5. All experimental procedures were approved by the CBMSO and Comunidad Autónoma de Madrid ethical committees.

### Human tissue
Paraffin sections of human embryonic eye primordia were provided by the Joint MRC/Wellcome Trust (grant# MR/R006237/1) Human Developmental Biology Resource (http://hdbr.org). Sections corresponded to samples CS13, -14, -15, and -16. CS staging allowed to determine the age of embryo as days post ovulation based on morphological landmarks (*O'Rahilly and Müller, 2010*).

### Generation of the Tg(E1-bhlhe40:GFP) line
Predictive enhancer and promoter epigenetic marks (*Bogdanovic et al., 2012*) were used to identify different potential regulatory elements of the *bhlhe40* gene (*Figure 1B*). Each region was amplified by PCR with specific primers (*Supplementary file 1*) and cloned using the pCR8/GW/TOPO TA Cloning Kit (Invitrogen). Plasmids were checked for enhancer insertion and the Gateway LR Clonase Enzyme Mix (Invitrogen) was used for recombination with the ZED vector (*Bessa et al., 2009*). The resulting constructs were injected together with Tol2 mRNA to generate the corresponding transgenic embryos, which were screened using a transgenesis efficiency marker present in the ZED vector (cardiac actin promoter:RFP). Positive larvae were grown to adulthood (F0) and then individually outcrossed with wild-type partners to identify founders. Founders were analysed using confocal microscopy. One of the lines corresponding to the enhancer E1 was finally selected and used for subsequent studies.

### Gal4-UAS-mediated expression
The *UAS:STMN1* construct was generated from the bidirectional UAS:GFP vector, which allows simultaneous and comparable production of GFP and the gene product of interest under the same regulatory sequences (*Paquet et al., 2009*; *Distel et al., 2010*). The gene was amplified by PCR using specific primers (*Supplementary file 1*) flanked by StuI restriction sites and the Expand High Fidelity PCR System, using the pQTEV-STMN1 (Addgene# 31326) construct as a mould. The PCR product was digested with StuI (Takara) and cloned into the pCS2 vector and thereafter isolated together with the polyA sequence of the vector by digestion with HindIII and SacII (Takara) and sub-cloned into the UAS:GFP plasmid. The generated plasmid (30 pg) was injected into the Tg(rx3:Gal4;UAS:RFP) (*Weiss et al., 2012*) line, together with Tol2 mRNA (50 pg) to increase efficiency.

### Embryos micro-injection and drug treatments
Embryos at one cell stage were injected using a Narishige micro-injector and glass needles prepared by horizontally pulling standard capillaries (filament, 1.0 mm, World Precision Instruments) with aP-97 Flaming/Brown Micropipette Puller (Sutter Instrument Company). A total of 30 pg for DNA and between 50 and 100 pg for mRNA in 1 nl volume were injected in the embryos in the cell or the yolk, respectively. Drug treatments were performed on manually dechorionated embryos at the desired developmental stage in E3 medium. The following compounds were used: blebbistatin (100 µM for 2.5 hr; Calbiochem); paranitroblebbistatin (20 µM; Optopharma), Ableb (5 µM for 15 min before photoactivation; Optopharma), and nocodazole (10 ng/µl for 2.5 hr; Sigma).

## In vitro transcription

The pCS2:Kaede, pCS2:EB3-GFP, and pCS2:H2B-RFP constructs were linearized and transcribed using the mMessage mMachine SP6 transcription kit (Invitrogen), following manufacturer's instructions. After transcription mRNAs were purified using the NucleoSpin RNA Clean-up kit (Machery Nagel).

In situ hybridization (ISH) *otx1* (previously known as *otx1b*) and *mitfa* probes were gifts from Prof. Steve Wilson (UCL, London, UK). The *bhlhe40* probe was generated by PCR from 24 hpf cDNA with specific primers *Supplementary file 1* using the Expand High Fidelity PCR System. Reverse primers included the T3 promoter sequence to in vitro transcribe the PCR product. In vitro transcription was performed using T3 RNA polymerase and DIG RNA labelling Mix (Roche) following manufacturer's instructions. Transcription products were precipitated with LiCl 0.4 M and 3 volumes of ethanol 100 % overnight at –20 °C. Samples were centrifuged at 4 °C and 12,000 *g* for 30 min, washed with ethanol 70 %, and re-suspended in 15 µl of RNAse-free water and 15 µl Ultra-Pure Formamide (Panreac). ISH were performed as described (*Cardozo et al., 2014*).

## BrdU incorporation assays

BrdU (Roche) was re-suspended in DMSO (Sigma) to generate stocks of 50 mg/ml that were kept at –20 °C. For Tg(E1-*bhlhe40*:GFP) zebra- and wild-type medaka fish groups of 15 embryos of stages comprised between 16 ss and 48 hpf were dechorionated and placed in BrdU solution (5 mg/ml in E3 medium) for 30 min on ice and then washed with fresh E3 medium. Embryos were let recover at 28 °C for 10 min before fixation in paraformaldehyde (PFA) 4 % overnight at 4 °C. For analysis in chick, BrdU (50 mg/egg) was added to each embryo 30 min before fixation. For analysis in mouse, pregnant dams were injected intraperitoneally with BrdU (50 µg/g), sacrificed 1 hr later and fixed. Chick and mouse embryos were immersion fixed in 4 % PFA in 0.1 M phosphate buffer, pH 7 at 4 °C for 4 hr and then washed in PBS and cryoprotected in 15% and 30% saccharose in 0.1 M phosphate buffer . All embryos were cryo-sectioned and the sections hydrated with PBS 1 X during 5 min and incubated in HCl during 40 min at 37 °C. After HCl treatment, sections were rinsed with PBS 1 X 10 times, and then processed for immunofluorescence as described below. The percentage of RPE proliferating progenitors was determined as the proportion of BrdU-positive cells over the total number of GFP (for E1-*bhlhe40*:GFP) or Otx2/Hoechst (medaka fish, chick, mouse embryos) positive cells in the RPE layer in each section. A minimum of three embryos and sections per embryo were counted (both eyes).

## Immunofluorescence

Zebrafish embryos at the corresponding stage for each experiment were fixed with 4 % (wt/vol) PFA (Merck) in 0.1 M phosphate buffer overnight at 4 °C. Whole-mount immunofluorescence was performed as described (*Cardozo et al., 2014*). Alternatively, embryos were incubated in 15 % sucrose – PBS overnight at 4 °C, embedded in 7.5 % gelatine (Sigma) 15 % sucrose (Merck), frozen in isopentane (PanReac) between –30°C and –40 °C and kept at –80 °C. Cryo-sectioning was performed with a cryo-stat (Leica CM 1950) at 20 µm thickness and dried overnight at room temperature. Chick and mouse embryos were collected, fixed 4 % PFA, equilibrated in sucrose, and cryo-sectioned as above. Paraffin sections of human embryonic tissue were de-paraffinized, washed in PBS, processed for antigen retrieval (10 mM citrate buffer, pH6, for 5 min at 110 °C in a boiling chamber, Biocaremedical), and subsequently processed together with all other samples for immunofluorescence. Immunostaining was performed as described (*Cardozo et al., 2014*) using the following primary antibodies: mouse anti-BrdU (1:200; Becton-Dickinson); chick anti-GFP (1:2000; Abcam); mouse anti-βcatenin (1:400, BD Transduction Laboratories); mouse anti-ZO-1 (1:400, Invitrogen); rabbit anti-laminin (1:200, Sigma); rabbit anti-Otx2 antibodies (1:1000; Abcam); rabbit anti-Ki67 (1:500, Abcam). The used secondary antibodies were conjugated with Alexa-488, Alexa-594, or Alexa-647 (1:500; Thermo Fisher). Sections were counterstained with Hoechst (Invitrogen), mounted in Mowiol, and analysed by conventional and confocal microscopy.

## Kaede photoconversion

Wild-type embryos were injected with Kaede mRNA. Embryos at 15 hpf with homogeneous green fluorescence were selected, mounted, and visualized under the Nikon AR1+ Confocal Microscope using a 20 ×/0.75 Plan-Apochromat objective. A region of interest (ROI) was drawn in the outer layer, corresponding to the putative position of the RPE progenitors, at a specific z-position and irradiated

with the 405 nm laser at 21 % of power for 10 loops to switch Kaede emission from green to red fluorescence. Due to confocality, photoconversion occasionally extended further than the selected plane, so that the tissues present above or below (i.e. ectoderm) also underwent photoconversion. After photoconversion embryos were let develop up to approximately 30 hpf stage, fixed and analysed by confocal microscopy for red fluorescence distribution.

## Ableb photoactivation

Ableb (*Kepiro et al., 2012*) was photoactivated with a Zeiss LSM 780 Upright multiphoton FLIM system with a W Plan-Apochromat 20 ×/1.0 DIC M27 75 mm WD 1.8 mm dipping objective. For each eye a specific ROI was drawn including RPE cells identified by GFP fluorescence. Ableb was activated in the ROIs using 860 nm wavelength and 20 mW laser power (this corresponds to 9–14 µW/µm$^2$ inside the ROI).

## Confocal imaging

Embryos were mounted with the appropriate orientation in 1.5 % low melting point agarose (Conda) diluted in E3 medium (for in vivo recording) or PBS (for fixed samples). Images were acquired either with a Nikon A1*R* + High Definition Resonant Scanning Confocal Microscope connected to an Inverted Eclipse Ti-E Microscope (20 ×/0.75 Plan-Apochromat, 40 ×/1.3 oil Plan-Fluor and 60 ×/1.4 oil Plan-Apocromat objectives) or with a Zeiss LSM710 Confocal Laser Scanning Microscope connected to a Vertical AxioImager M2 Microscope (40 ×/1.3 oil Plan-Apochromat, W N-Achroplan 20 ×/0.5, W Plan-Apochromat 40 ×/1.0 DIC VIS-IR).

## 3D reconstructions

3D videos (i.e. *Figure 1—videos 1–3*) were generated from full stacks using the 3D project option in Fiji (*Schindelin et al., 2012*). RPE surface renderings were generated using Imaris (Bitplane), with a value of 6 in Surface Area Detail and 7 in Background Subtraction.

## Morphometric analysis

Unless otherwise specified, morphometric analysis of cells and tissues was performed using Matlab (The Mathworks, Natick, MA) using the XYZ coordinates of the processed images or Fiji (*Schindelin et al., 2012*). This analysis was performed using previously processed fluorescent images from videos of Tg(E1-*bhlhe40*:GFP; *rx3*:GAL4;UAS:RFP) or Tg(E1-*bhlhe40*:GFP) and H2B-RFP-injected embryos (*Figure 1—video 2* and *Figure 3—video 1*), from which the signal corresponding to the RPE or the whole OV/OC were isolated semi-manually with the help of Fiji macros and tools designed to select 3D structures. The RPE-specific GFP signal was processed with a median filter. In the case of *Figure 3—video 1*, the background ramp for the GFP signal was neutralized in each frame via subtraction of a copy of itself after a grey-scale morphological operation (*Hassanpour et al., 2015*; *Arce, 2005*). For all videos, the median intensity was thereafter established as the cutoff value for differentiating background and signal (i.e. pixel with an intensity lower than the cutoff were set to zero) for all images that were in both videos. The signal derived from H2B was localized in cell nuclei, and therefore it was post-processed with a grey-scale closing operation to fill empty spaces between nuclei. Morphometric analysis was performed in the resulting processed images. All values were calculated in microns by scaling the x, y, z coordinates according to the following: (0.62 µm × 0.62 µm × 1.37 µm) for *Figure 1—video 2* and (0.62 µm × 0.62 µm × 1.07 µm) for *Figure 3—video 1*. Volumes (µm$^3$) were calculated as the number of voxels with a value higher than 0. RPE surface (µm$^2$) was calculated applying a second-order linear adjustment on the plane YZ corresponding to the plane of the OV/OC hinges with the fit function available in Matlab (The Mathworks, Natick, MA). RPE thickness (µm) was determined as the result of volume (µm$^3$)/surface (µm$^2$). Unfortunately, semi-manual RPE image extraction was not perfect, when GFP signal associated to CMZ development arises. To account for this problem, the GFP signal for each frame was divided into seven equivalent blocks using the x, y coordinates from the z-projection of each frame. In this case, RPE volume and surface were calculated independently in each one of the regions up to 20 hpf, when the most anterior block (now corresponding to the arising CMZ) was discarded from the analysis. For the subsequent frames the two anterior most blocks were discarded (*Figure 3—figure supplement 1*). The total OV/OC volume (µm$^3$) was determined using the red fluorescence from the Tg(rx3:GAL4;UAS:RFP) embryos at

17–22hpf. H2B expression was used to determine the volume of the OC (H2B volume in *Figure 3*) as follows for each frame of *Figure 3—video 1*: the maximum, Gaussian blur and minimum filters were applied to the image; subsequently, the convex hull (*Hamburger and Hamilton, 1992*) was calculated for the image to obtain the geometrical shape that covers all pixels with an intensity higher than 0, including the lens; finally, only the regions present in the image and the convex hull are used to define the H2B volume. Individual cell area was determined in cells located at a medial position of the OV for each cell type (progenitor, RPE, and NR); cell contour was drawn using the segmented line tool in Fiji (*Schindelin et al., 2012*). Apico-basal (A-B) length (μm) of individual cells was estimated by manually tracing a line from the basal to the apical membrane in the z-position in which the nucleus had its larger surface using the straight-line tool in Fiji (*Schindelin et al., 2012*). To account for possible developmental asynchrony when eyes from the same embryo were differentially treated (irradiated vs. non-irradiated), the A-B length of the irradiated eye was normalized with that of the non-irradiated eye. Values above 1 indicated less RPE cell flattening in experimental eyes. The invagination angle was determined as previously described (*Sidhaye and Norden, 2017*) using manual drawing with the Fiji angle tool (*Schindelin et al., 2012*). The vertex of the angle was placed approximately in the centre of the basal surface of the NR and the vectors were drawn up to the edges of the CMZ. Angles were measured in the z-positions in which the irradiated RPE was maximally affected and compared to equivalent positions of control non-irradiated eyes. Values were normalized with those of the contralateral non-treated eye, to account for possible asynchronies.

## Statistical analysis

All statistical analysis was performed with IBM SPSS Statistics version 20.0. The method used is indicated in each case together with the sample size.

## Acknowledgements

We wish to thank Drs JR Martinez-Morales and E Marti for critical reading of the manuscript; the confocal microscopy service of the CBMSO and CNIC (Centro Nacional de Investigaciones Cardiovasculares) for help with image acquisition and analysis and the fish facilities for caring of the zebrafish lines. This work was supported by grants from the Spanish AEI (BFU2014-55918-P to FC; BFU2016-75412-R with FEDER support, RED2018-102553-T and PID2019-104186RB-100 to PB), BBVA Foundation (N[16]_BBM_BAS_0078 to FC) and Fundación Ramon Areces-2016 (to PB). TMM and ML were supported by FPU (FPU14/02867) and FPI (BES-2015–073253) pre-doctoral contracts from the Spanish AEI, respectively. We also acknowledge a CBM Institutional grant from the Fundación Ramon Areces.

## Additional information

### Funding

| Funder | Grant reference number | Author |
| --- | --- | --- |
| Agencia Estatal de Investigación | PID2019-104186RB-100 | Paola Bovolenta |
| Ministerio de Economía, Industria y Competitividad, Gobierno de España | RED2018-102553-T | Paola Bovolenta |
| Ministerio de Economía, Industria y Competitividad, Gobierno de España | BFU2016-75412-R | Paola Bovolenta |
| Ministerio de Economía, Industria y Competitividad, Gobierno de España | BFU2014-55918-P | Florencia Cavodeassi |
| BBVA Foundation | N[16]_BBM_BAS_0078 | Florencia Cavodeassi |
| Fundación Ramon Areces-2016 | | Paola Bovolenta |

| Funder | Grant reference number | Author |
|---|---|---|

The funders had no role in study design, data collection and interpretation, or the decision to submit the work for publication.

## Author contributions

Tania Moreno-Mármol, Conceptualization, Data curation, Formal analysis, Investigation, Methodology, Visualization, Writing – original draft; Mario Ledesma-Terrón, Formal analysis, Investigation, Methodology; Noemi Tabanera, Maria Jesús Martin-Bermejo, Investigation, Methodology; Marcos J Cardozo, Investigation, Visualization; Florencia Cavodeassi, Conceptualization, Funding acquisition, Supervision, Writing – original draft; Paola Bovolenta, Conceptualization, Funding acquisition, Supervision, Writing - review and editing

## Author ORCIDs

Florencia Cavodeassi (iD) http://orcid.org/0000-0003-4609-6258
Paola Bovolenta (iD) http://orcid.org/0000-0002-1870-751X

## Ethics

Complying with EU regulations.

## Decision letter and Author response

Decision letter https://doi.org/10.7554/eLife.63396.sa1
Author response https://doi.org/10.7554/eLife.63396.sa2

# Additional files

## Supplementary files

• Transparent reporting form
• Supplementary file 1. List of primers used in this study.

## Data availability

All data generated or analysed during this study are included in the manuscript and supporting files. Source data files have been provided for all the graphs shown in the study.

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
