## [Decision Letter]

**Acceptance summary:**

The authors describe a novel tool [Tg(E1-bhlhe40:GFP)] that can be used to monitor the earliest phases of zebrafish retinal pigment epithelium (RPE) development. The results demonstrate the RPE cell flattening in zebrafish occurs independent of the basal constriction / apical expansion of neural retina progenitor cells and that this process is affected by perturbation of actomyosin activity or microtubule organization. This study provides a deeper understanding of RPE and optic cup morphogenesis.

**Decision letter after peer review:**

Thank you for submitting your article "Stretching of the retinal pigment epithelium contributes to zebrafish optic cup morphogenesis" for consideration by *eLife*. Your article has been reviewed by 2 peer reviewers, and the evaluation has been overseen by Marianne Bronner as the Senior Editor and Reviewing Editor. The reviewers have opted to remain anonymous.

The reviewers have discussed the reviews with one another and the Reviewing Editor has drafted this decision to help you prepare a revised submission.

We would like to draw your attention to changes in our revision policy that we have made in response to COVID-19 (https://elifesciences.org/articles/57162). Specifically, when editors judge that a submitted work as a whole belongs in eLife but that some conclusions require a modest amount of additional new data, as they do with your paper, we are asking that the manuscript be revised to either limit claims to those supported by data in hand, or to explicitly state that the relevant conclusions require additional supporting data.

Our expectation is that the authors will eventually carry out the additional experiments and report on how they affect the relevant conclusions either in a preprint on bioRxiv or medRxiv, or if appropriate, as a Research Advance in eLife, either of which would be linked to the original paper.

Summary:

Moreno-Marmol and colleagues describe the general morphogenic rearrangements of zebrafish RPE cells during optic cup formation, providing novel insights to the autonomous nature of RPE cell flattening from the changes occurring in the neural retinal progenitors. They postulate that in fast developing species, RPE progenitor cells stretch to cover the optic vesicle and aid in its development whereas in slower developing species, RPE progenitors undergo more extensive and prolonged proliferation to provide the cell coverage to envelop the optic cup. Both reviewers see considerable merit in the work and find the data interesting and thought-provoking. However, they also suggest some important and feasible modifications. These are described in more detail in the full reviews which are appended below.*Reviewer #1:*

In their study, Moreno-Marmol and colleagues describe the general morphogenic rearrangements of zebrafish RPE cells during optic cup formation. Novel insights to the autonomous nature of RPE cell flattening from the changes occurring in the neural retinal progenitors is presented. Through comparative analysis of RPE progenitor cell proliferation, the Authors put forward and provide data that supports the hypothesis that in fast developing species, RPE progenitor cells stretch to cover the optic vesicle and aid in its development. In slower developing species, RPE progenitors undergo more extensive and prolonged proliferation to provide the cell coverage needed to envelop the optic cup.

Overall the paper is well written and addresses questions that will garner the attention of those interested in ocular development. The authors describe a novel tool [Tg(E1-bhlhe40:GFP)] that can be used to monitor the earliest phases of zebrafish RPE development. There is lost opportunity however, in that neither Gal4 nor Cre / CreERT2 lines were not also developed. This would have been useful in better parsing the interactions between RPE and NR progenitors. Still, experiments described in their paper convincingly demonstrate the RPE cell flattening in zebrafish occurs independent of the basal constriction / apical expansion of the NR progenitors cells. They also convincingly show that this process is affected when either the actomyosin activity or microtubule organization is perturbed. These are outstanding starting observations to provide a deeper understanding of RPE and optic cup morphogenesis, but given the limited scope of the analysis, this work is better suited for a more specialized journal. Providing more in depth analysis of E1-bhlhe40 activation and conservation across species, or more detailed mechanisms into the cytoskeletal regulation and reorganization required for RPE cell flattening is warranted (such as those alluded to from work in the fly amnioserosa).

Substantive Concerns:

1. In Figure 3B, it was unclear (to me) the utility of the H2B construct, and how the H2B volume was calculated (was total nuclear volume inferred?)

2. For the experiment described in Figure 4, inactivation of myosinII (ABleb irradiation) appears to have a stronger effect on RPE in the ventral part of the eye. Given the spot irradiation occurred at the region of the eye-bud specified to give rise to RPE (Figure 1E), could 'pinwheeling' migration of the RPE progenitors contribute the phenotype? Inactiviation of actomyosin function at several stages might be informative.

3. With regard to use of the bidirectional UAS constructs that facilitated expression of STMN1 or CCND1, how faithful is the concordance of GFP and STMN1 or CCND1 protein? Similarly, is it possible that low expressing cells (faint or undetectable GFP), still produce enough STMN1 or CCND1 protein to affect microtubule network / proliferation, even if stoichiometrically produced with GFP? This is particularly important given the use of a the rx3 promoter which expresses in both RPE and NR progenitors. Finally, given the migration of RPE cells, those that are isolated from non-expressing NR may have shortly before been quite proximal to NR that did express STMN1. A more elegant approach to distinguish the autonomous from non-autonomous effects would be to use promoters more specific to each cell type (such as the E1-bhlhe40 element for RPE).*Reviewer #2:*

In the present work, Moreno-Mármol and colleagues investigate the morphological changes of RPE cells during optic cup formation in zebrafish. Using a novel transgenic line that allows them to follow the process in 4D and from earlier stages than previously available lines, they show that RPE cells in zebrafish stretch during optic vesicle folding. Following a number of elegant approaches, they show an autonomous process happening in the RPE that accompanies the NR folding, and speculate that RPE flattening is an evolutionary adaptation due to the fast dynamics of the process. Finally they show comparative data using other vertebrates that develop slower – in what is perhaps the highest point of the manuscript – revealing a different strategy that involves a sustained proliferation of RPE cells; the additional number of cells, therefore, allows for the RPE covering the folding NR without the need of such a massive stretching.

I have some points that I would like the authors to address.

#1 It is clear from their results that flattening can occur in RPE cells regardless of OC folding (Figure 4J). But is it a driving force for the OC folding? I was left with the impression that the authors are ambiguous about it. Their interpretation from figure 5F is that retina can still fold in the absence of RPE flattening (although the OC folding is far from a control case). In 5J, however, they conclude that flattening is indeed affecting NR invagination. A unifying quantification of the NR folding (as presented in Figure 4) would help assessing the role of RPE flattening as a driving force for OC folding in zebrafish. In general, I would like to see a clearer statement on the co-occurence of both processes in the RPE and NR. Are both of them tissue-autonomous? Are they independent of each other? Or is there some degree of inter-dependance, and at what level does it happen?

#2 Section "RPE flattening is a cell autonomous process required for proper OV folding". The authors show that RPE cell flattening is a tissue autonomous process, not a cell autonomous process. The experiments illustrated in Figure 4 report an all-or-non response in the RPE. Form the experiments shown on lines 275-276 (UAS:stmn1), it seems that the flattening effect is non cell autonomous. In fact, the authors use during the discussion "tissue autonomous", which is more accurate. The authors should either demonstrate cell autonomy by generating a mosaic RPE in the experiments of figure 4, or change their statement to tissue autonomy. If they go for tissue autonomy (as their experiments demonstrate), please discuss about the non-cell autonomy of the flattening response.

#3 The experiments using UAS:stmn1 shown that, although the expression is clonal and targets only a few RPE cells, the non-flattening effects involved pretty much the entire RPE (Figure 5H). Is the RPE responding as one unit? Along the manuscript there is no case in which the RPE responds in a mosaic manner. Why is that? The authors leave this point unexplored. If RPE cells are coordinated in their response, how does it happen? Any hint in this regard will help framing the results in a better context.

#4 The data obtained by using UAS:ccnd1 contrasts with the quality of the other results shown along the manuscript. It is not clear to me whether the forced expression of ccnd1 is indeed promoting proliferation in this system (the authors should provide this data to validate the tool), nor whether the reported differences (52.5% vs 47.5%) are significant or not. They seem too small differences to be responsible of the reported phenotype. Given the FO mosaic approach implemented, the authors could rapidly assess more candidates to drive sustained proliferation in RPE cells. Being the last part of the manuscript about other species that resolve the same problem in a different manner – i.e., adding more cells rather than stretching the existing ones – I find this to be a critical point that needs to be shown clearly.

[Editors' note: further revisions were suggested prior to acceptance, as described below.]

Thank you for resubmitting your article "Stretching of the retinal pigment epithelium contributes to zebrafish optic cup morphogenesis" for consideration by *eLife*. Your revised article has been reviewed by 2 peer reviewers, and the evaluation has been overseen by Marianne Bronner as the Senior Editor and Reviewing Editor. The reviewers have opted to remain anonymous.

The reviewers have discussed the reviews with one another and the Reviewing Editor has drafted this decision to help you prepare a revised submission.

While the reviewers are largely satisfied, one remaining revision is (please see complete review below).

Essential revisions:

1. Reviewer 3 raises and important point concerning over-expression of ccnd1 in RPE cells – the point that sets the basis for the "proliferation-OR-stretching". This was weak in the previous version and has not improved during the revision and would require the generation of a new transgenic line which may already be in place.

2. Lines 320 to 330 are problematic, since they show only a weak trend (and not a "small increase" in proliferation, since the statistical analysis indicates that the differences are not significative).*Reviewer #1:*

Overall the paper is well written and addresses questions that will garner the attention of those interested in ocular development and epithelial morphogensis. The authors describe a novel tool [Tg(E1-bhlhe40:GFP)] that can be used to monitor the earliest phases of zebrafish RPE development. Ideally, some experiments would have been better performed using promoters that drive gene expression specifically in RPE and NR progenitors, as opposed to the rx3 promoter which expresses in both cell types. Still, experiments described in their paper convincingly demonstrate the RPE cell flattening in zebrafish occurs independent of the basal constriction / apical expansion of the NR progenitors cells. They also convincingly show that this process is affected when either the actomyosin activity or microtubule organization is perturbed. These are outstanding starting observations to provide a deeper understanding of RPE and optic cup morphogenesis. This study sets the stage for more in depth analysis of E1-bhlhe40 activation and conservation across species, as well as a more detailed mechanisms into the cytoskeletal regulation and reorganization required for RPE cell flattening.

The Authors have adequately addressed my concerns and the paper reads very nicely. Ideally, the cell type specific expression experiments would be included in this version, but experimental challenges and time to generate the new lines are appreciated. I highly encourage the Authors to follow through with updates on the planned experiments and the results as a "Research Advance" to be associated with this paper.*Reviewer #3:*

In the revised version of their manuscript, Moreno-Mármol and colleagues have addressed most of the main concerns I had raised. The authors have improved their previous versions by emphasising the main concept of the work and providing a better explanation for their hypothesis. Overall, the work now describes the inter-relation between the NR and the RPE during retinogenesis, highlighting that morphological changes occurring in the zebrafish RPE are critical for the retina morphogenesis. The authors report that species in which the RPE stretching does nor occur show an increase RP proliferation, suggesting two alternative ways to coordinate the grow of RPE and NR. The observation stays descriptive, since the attempts from the authors to switch stretching-to-proliferation modes in zebrafish were not successful – see recommendation for authors below.

One of my main points was to provide more accuracy in the "cell-autonomous" effect of RPE cells. The authors have now changed their statement to a "tissue-autonomous" effects, which in my view fits better the data they have acquired. The idea of the RPE behaving in a syncytial-like manner is very interesting indeed, and the discussion has been enriched by this addition.

The one point I do not find satisfactory addressed is the one concerning over-expression of ccnd1 in RPE cells – the point that sets the basis for the "proliferation-OR-stretching". This is a point that was raised as well by reviewer 1; it was weak in the previous version and has not improved during the revision. I consider the experiment to be critical, in light of the exciting results the authors provide for different species. The generation of a new transgenic line is an enterprise that can take longer that the time allowed for a revision, but I understand that the authors already have the tools to go for it? ("… a new construct we have developed"). Furthermore, the mosaic approach followed by the authors in their initial submission allows the analysis in the injected generation and therefore within days, so I would have expected this point to be tackled experimentally. In their current form, the lines 320 to 330 are the most problematic of the entire manuscript, since they show only a weak trend (and not a "small increase" in proliferation, since the statistical analysis indicates that the differences are not significative). Presenting additional data in a subsequent "Research Advance" seems adequate for other experiments required by reviewer 1 and myself, but I feel this point is central to the story and I can not picture major problems in performing transient experiments to strengthen the proposed statement – keeping the proliferative state impairs RPE flattening. Is it realistic that the authors perform a transient over-expression of ccnd1 using an RPE specific promoter? – a construct they already have. If the authors are experiencing restrictions to the lab due to the pandemic and it is not possible to perform additional experiments, then I would suggest removing the lines 320 to 330. The manuscript would miss an interesting, important aspect but will gain in accuracy.

I would strongly encourage the authors to tackle this point in the present version of the manuscript, though. If so, I would ask the authors to consider reporting this dataset at the end of the manuscript – after the report on the difference species in Figure 7. In this manner, they could show: (a) first, the inverse correlation between flattening and proliferation across vertebrates, (b) then prove that in one model species the switch can be induced experimentally.

---

## [Author Response]

Reviewer #1:In their study, Moreno-Marmol and colleagues describe the general morphogenic rearrangements of zebrafish RPE cells during optic cup formation. Novel insights to the autonomous nature of RPE cell flattening from the changes occurring in the neural retinal progenitors is presented. Through comparative analysis of RPE progenitor cell proliferation, the Authors put forward and provide data that supports the hypothesis that in fast developing species, RPE progenitor cells stretch to cover the optic vesicle and aid in its development. In slower developing species, RPE progenitors undergo more extensive and prolonged proliferation to provide the cell coverage needed to envelop the optic cup.Overall the paper is well written and addresses questions that will garner the attention of those interested in ocular development. The authors describe a novel tool [Tg(E1-bhlhe40:GFP)] that can be used to monitor the earliest phases of zebrafish RPE development. There is lost opportunity however, in that neither Gal4 nor Cre / CreERT2 lines were not also developed. This would have been useful in better parsing the interactions between RPE and NR progenitors. Still, experiments described in their paper convincingly demonstrate the RPE cell flattening in zebrafish occurs independent of the basal constriction / apical expansion of the NR progenitors cells. They also convincingly show that this process is affected when either the actomyosin activity or microtubule organization is perturbed. These are outstanding starting observations to provide a deeper understanding of RPE and optic cup morphogenesis, but given the limited scope of the analysis, this work is better suited for a more specialized journal. Providing more in depth analysis of E1-bhlhe40 activation and conservation across species, or more detailed mechanisms into the cytoskeletal regulation and reorganization required for RPE cell flattening is warranted (such as those alluded to from work in the fly amnioserosa).

We agree with the reviewer that generating a Gal4/Cre line based on the *bhlhe40* enhancer would have been a great addition to our toolkit and in fact we did attempt to generate such a line. Unfortunately, the result was not satisfactory, as the line did not show the expected specificity, in retrospect, likely due to a problem of insulators. This is why we resorted to the use of the rx3:Gal4-line. To overcome such a problem, we have now taken a different approach and modified the original plasmid that we used to generate the Tg(E1-*bhlhe40*:GFP) line so that now we can use it to overexpress any gene of interest in the RPE. We have already verified that the new construct faithfully reproduces the pattern of the Tg(E1-*bhlhe40*:GFP), and we have further used it to generate new transgenic lines. We will exploit this construct to force STMN1 and *ccnd1* expression in the RPE as detailed below. However, these experiments will require several months and therefore, following the indication of the Editor, we will report them as a “Research Advances” later on, linking the obtained results to this manuscript.

Substantive Concerns:1. In Figure 3B, it was unclear (to me) the utility of the H2B construct, and how the H2B volume was calculated (was total nuclear volume inferred?)

The RFP reporter expression of the *rx3* line used at early stages is less visible later on. We therefore used H2B to counterstain the optic primordium. Although a cytoplasmic marker would have avoided the process of artificially “filling the gaps”, H2B provided a bright and reliable signal. The methods used to fill the gaps and infer the volume of the whole eye primordium are explained in the “morphometric analysis” section of the methods section together with the procedures used to quantify the entire eye volume. The approach that we took has been carefully designed and does not affect the rigor of the analysis. We have now re-written and clarified this description (page 21-22).

2. For the experiment described in Figure 4, inactivation of myosinII (ABleb irradiation) appears to have a stronger effect on RPE in the ventral part of the eye. Given the spot irradiation occurred at the region of the eye-bud specified to give rise to RPE (Figure 1E), could 'pinwheeling' migration of the RPE progenitors contribute the phenotype? Inactiviation of actomyosin function at several stages might be informative.

Indeed, in the ABleb experiment the most severe alterations were observed in the RPE cells closer to the ventral region. As we now indicate in the text (page 8) this is likely because cells flatten and spread in the medio-lateral direction, exactly following the “pinwheel movement” described by Kwan et al., (2012 Development) -to which we believe the reviewer refers to. We cannot exclude that cell movements do exist and contribute to the localization of the cells ventrally. However, RPE cells spreading may give the impression of “migration”.

The reviewer also suggests interfering with actomyosin at several stages. There is a technical and a conceptual reason why we have not done so. We cannot assess the effect at earlier stages as we rely on the onset of expression of E1-*bhlhe40*:GFP to identify RPE cells, whereas at later stages the optic vesicle is already folded and the RPE already flattened, making it very challenging to irradiate exclusively RPE cells and in sufficient number. Besides, the question we addressed in our manuscript focuses on the contribution of RPE morphogenesis to OV folding. We expect that once the optic cup is formed, interference with RPE cytoskeleton will have only a minor impact on the overall shape of the eye.

3. With regard to use of the bidirectional UAS constructs that facilitated expression of STMN1 or CCND1, how faithful is the concordance of GFP and STMN1 or CCND1 protein?

The bidirectional UAS construct we have used was originally described in Paquet et al., 2009, J Clin Inv. 119, 1382–1395 and then further used in Distel et al., 2010, J Cell Biol. 191, 875–90 and Distel et al., 2011, Commun Integr Biol. 4, 336-9. In the original publication, Paquet et al., show that the bidirectional UAS efficiently express both of the tested proteins at similar levels, although they did not perform an explicit stoichiometric analysis. This vector has been successfully used in several additional publications (see https://zfin.org/ZDB-TGCONSTRCT-111028-1 for a list). We are thus confident that there is concordance between GFP and either *STMN1* or *ccnd1* expression. To clarify this point we now mention how the vector works in the corresponding section in methods (page 18) and in results (page 8). We also added the reference of Paquet et al., not included in our previous version.

Similarly, is it possible that low expressing cells (faint or undetectable GFP), still produce enough STMN1 or CCND1 protein to affect microtubule network / proliferation, even if stoichiometrically produced with GFP? This is particularly important given the use of a the rx3 promoter which expresses in both RPE and NR progenitors. Finally, given the migration of RPE cells, those that are isolated from non-expressing NR may have shortly before been quite proximal to NR that did express STMN1. A more elegant approach to distinguish the autonomous from non-autonomous effects would be to use promoters more specific to each cell type (such as the E1-bhlhe40 element for RPE).

We fully agree. As mentioned in answer to point 1, this was our original, although so far failed, plan. As also mentioned, we will perform cell specific studies using a modified ZED-E1-*bhlhe40* plasmid. These experiments however will require some time and we have therefore decided to follow the editor’s recommendation and incorporate the new data to this study in the form of an associated Research Advance report.

Reviewer #2:In the present work, Moreno-Mármol and colleagues investigate the morphological changes of RPE cells during optic cup formation in zebrafish. Using a novel transgenic line that allows them to follow the process in 4D and from earlier stages than previously available lines, they show that RPE cells in zebrafish stretch during optic vesicle folding. Following a number of elegant approaches, they show an autonomous process happening in the RPE that accompanies the NR folding, and speculate that RPE flattening is an evolutionary adaptation due to the fast dynamics of the process. Finally they show comparative data using other vertebrates that develop slower – in what is perhaps the highest point of the manuscript – revealing a different strategy that involves a sustained proliferation of RPE cells; the additional number of cells, therefore, allows for the RPE covering the folding NR without the need of such a massive stretching.I have some points that I would like the authors to address.#1 It is clear from their results that flattening can occur in RPE cells regardless of OC folding (Figure 4J). But is it a driving force for the OC folding? I was left with the impression that the authors are ambiguous about it. Their interpretation from figure 5F is that retina can still fold in the absence of RPE flattening (although the OC folding is far from a control case). In 5J, however, they conclude that flattening is indeed affecting NR invagination. A unifying quantification of the NR folding (as presented in Figure 4) would help assessing the role of RPE flattening as a driving force for OC folding in zebrafish. In general, I would like to see a clearer statement on the co-occurence of both processes in the RPE and NR. Are both of them tissue-autonomous? Are they independent of each other? Or is there some degree of inter-dependance, and at what level does it happen?

Indeed, we propose that the RPE is a driving force for OV folding. As represented in the model illustrated in Figure 8, the mechanical force generated by RPE stretching synergizes with those generated by rim involution and neural retina basal constriction to the overall OV folding. We apologize if this message was not explicit enough, although we believe that the heading in page 7 indeed stated this message that now reads “RPE flattening is a tissue autonomous process required for proper OV folding” (we changed cell for tissue). We have also edited a few additional sentences in the abstract, end of the introduction, results and discussion to make this message more explicit than what it was before.

#2 Section "RPE flattening is a cell autonomous process required for proper OV folding". The authors show that RPE cell flattening is a tissue autonomous process, not a cell autonomous process. The experiments illustrated in Figure 4 report an all-or-non response in the RPE. Form the experiments shown on lines 275-276 (UAS:stmn1), it seems that the flattening effect is non cell autonomous. In fact, the authors use during the discussion "tissue autonomous", which is more accurate. The authors should either demonstrate cell autonomy by generating a mosaic RPE in the experiments of figure 4, or change their statement to tissue autonomy. If they go for tissue autonomy (as their experiments demonstrate), please discuss about the non-cell autonomy of the flattening response.

We appreciate the reviewer remark. We believe that overall our studies call for “tissue autonomy” rather than individual cell autonomy. The activation of ABleb occurs in a limited number of cells but a larger number of cells seem to remain cuboidal in shape. A similar “community effect” is observed upon mosaic Gal4-mediated expression of *STMN1* or *ccnd1* in RPE cells. We apologize if this was not clearly explained in the text. As addressed in the point below, there is evidence in the literature indicating that RPE cells are interconnected and behave as a community. Following the reviewer’s advice, we have now changed the terminology throughout the text from cell-autonomous to tissue-autonomous.

#3 The experiments using UAS:stmn1 shown that, although the expression is clonal and targets only a few RPE cells, the non-flattening effects involved pretty much the entire RPE (Figure 5H). Is the RPE responding as one unit? Along the manuscript there is no case in which the RPE responds in a mosaic manner. Why is that? The authors leave this point unexplored. If RPE cells are coordinated in their response, how does it happen? Any hint in this regard will help framing the results in a better context.

The reviewer points to a very interesting aspect of RPE behavior that has caught our attention and indeed needs further investigation. However, we have not done so in this manuscript as we felt and still feel that asking whether the RPE behaves in “syncytial-like” manner was somewhat distracting from the main question of the present manuscript: does RPE stretching contribute to OV folding? Although at present we can just offer a speculation, we believe that the RPE indeed acts in a “syncytial-like” manner. This kind of behavior is for example observed among astrocytes, which, through a Gap-43 mediated connections, can propagate Ca^++^ waves in the brain (see, Buskila et al., 2019, Front. Neurosci. 13, 1125; for a review). In unpublished results, we have observed that Gap-43 is one of the earliest markers of RPE cells in different species. This expression is maintained at adult stages when gap junctions connecting RPE cells have been described (Kerr et al., IVOS 2010; Hoang et al., 2010, Mol Vis). Likely these connections are responsible of the described chemically coupling observed in RPE cells and the propagation of Ca^++^ waves, as observed in astrocytes (Pearson et al., 2004, Eur J Neurosci; Bao et al., 2019, Invest Ophthalmol Vis Sci). In addition, in a parallel study (Buono et al., 2020, in the ref list), we have shown that st18 RPE cells already express many desmosomal proteins, indicating that the cells establish tight connections very soon. These connections promote cohesion that in other contexts, as for example zebrafish epiboly (i.e Song et al., 2013, Dev Cell 24: 486–501; Petridou et al., 2019 Nat Cell Biol 21: 169–178; Camacho-Macorra et al., 2020, bioRxiv doi.org/10.1101/2020.12.01.407478), are known to promote coordinated cell spreading. Such a coordinated behavior seems to occur also in the RPE. Finally, it might be worth mentioning that the existence of “a community effect” during embryonic development has been described decades ago for different tissues, including the eye (Gurdon, 1988, Nature, 336, 772-4). We have included some of these ideas in the discussion to provide an answer to the reviewer questions (page 14).

#4 The data obtained by using UAS:ccnd1 contrasts with the quality of the other results shown along the manuscript. It is not clear to me whether the forced expression of ccnd1 is indeed promoting proliferation in this system (the authors should provide this data to validate the tool), nor whether the reported differences (52.5% vs 47.5%) are significant or not. They seem too small differences to be responsible of the reported phenotype. Given the FO mosaic approach implemented, the authors could rapidly assess more candidates to drive sustained proliferation in RPE cells. Being the last part of the manuscript about other species that resolve the same problem in a different manner – i.e., adding more cells rather than stretching the existing ones – I find this to be a critical point that needs to be shown clearly.

We agree with the reviewer. As mentioned in our answer to reviewer 1 (point 1) we will repeat this experiment using RPE specific overexpression of *ccnd1* with a new construct we have developed. With this approach we will also use different genes that may force RPE proliferation. However, as we mentioned the results of these experiments will be reported later as a Research advance. Nevertheless, the use of the UAS:*ccnd1* construct shows that its forced expression prevents cell flattening and interferes *bhlhe40* expression, although the data related to forced proliferation are weak with no statistical significance (now indicated in the text), likely due to the limited number of embryos we could analyze.

[Editors' note: further revisions were suggested prior to acceptance, as described below.]

Essential revisions:1. Reviewer 3 raises and important point concerning over-expression of ccnd1 in RPE cells – the point that sets the basis for the "proliferation-OR-stretching". This was weak in the previous version and has not improved during the revision and would require the generation of a new transgenic line which may already be in place.

We do agree that this point is weak. As we fully explain below in our answer to reviewer 3, our attempt to generate a new transgenic line has failed. The reasons are unclear and we can only put forward a few hypotheses as we explain below.

2. Lines 320 to 330 are problematic, since they show only a weak trend (and not a "small increase" in proliferation, since the statistical analysis indicates that the differences are not significative).

We agree with this appreciation and given that we have been unable to force proliferation in the prospective RPE (see answer to reviewer 3 for more information), we have removed these lines from the text, together with the corresponding panels and legends in Figure 6. We have also deleted the methods related to the generation of the *UAS:ccdn1* construct.

In addition to the changes mentioned above we have revised the text at our best to catch all typos, as indicated by reviewer 3, and improve the readability of a few sentences. We have also expanded our description of the work reported in ref 22 as it supports well the notion that proliferation has a minimal contribution to RPE expansion in zebrafish (page 10, bottom and top of page 14). We apologize for not having included

this information in the first version, as the data escaped our attention.

Reviewer #1:Overall the paper is well written and addresses questions that will garner the attention of those interested in ocular development and epithelial morphogensis. The authors describe a novel tool [Tg(E1-bhlhe40:GFP)] that can be used to monitor the earliest phases of zebrafish RPE development. Ideally, some experiments would have been better performed using promoters that drive gene expression specifically in RPE and NR progenitors, as opposed to the rx3 promoter which expresses in both cell types. Still, experiments described in their paper convincingly demonstrate the RPE cell flattening in zebrafish occurs independent of the basal constriction / apical expansion of the NR progenitors cells. They also convincingly show that this process is affected when either the actomyosin activity or microtubule organization is perturbed. These are outstanding starting observations to provide a deeper understanding of RPE and optic cup morphogenesis. This study sets the stage for more in depth analysis of E1-bhlhe40 activation and conservation across species, as well as a more detailed mechanisms into the cytoskeletal regulation and reorganization required for RPE cell flattening.The Authors have adequately addressed my concerns and the paper reads very nicely. Ideally, the cell type specific expression experiments would be included in this version, but experimental challenges and time to generate the new lines are appreciated. I highly encourage the Authors to follow through with updates on the planned experiments and the results as a "Research Advance" to be associated with this paper.

We thank the reviewer for the comments and her/his appreciation of the experimental challenges associated with the generation of new zebrafish line. As we explain below in our answer to reviewer 3, so far, we have been unable to generate new lines limiting the expression of specific genes to the RPE. In particular, we failed to force proliferation in the RPE, likely because this tissue differentiates “en bloc”, as we suggest in our manuscript (see also below).

Reviewer #3:In the revised version of their manuscript, Moreno-Mármol and colleagues have addressed most of the main concerns I had raised. The authors have improved their previous versions by emphasising the main concept of the work and providing a better explanation for their hypothesis. Overall, the work now describes the inter-relation between the NR and the RPE during retinogenesis, highlighting that morphological changes occurring in the zebrafish RPE are critical for the retina morphogenesis. The authors report that species in which the RPE stretching does nor occur show an increase RP proliferation, suggesting two alternative ways to coordinate the grow of RPE and NR. The observation stays descriptive, since the attempts from the authors to switch stretching-to-proliferation modes in zebrafish were not successful – see recommendation for authors below.One of my main points was to provide more accuracy in the "cell-autonomous" effect of RPE cells. The authors have now changed their statement to a "tissue-autonomous" effects, which in my view fits better the data they have acquired. The idea of the RPE behaving in a syncytial-like manner is very interesting indeed, and the discussion has been enriched by this addition.The one point I do not find satisfactory addressed is the one concerning over-expression of ccnd1 in RPE cells – the point that sets the basis for the "proliferation-OR-stretching". This is a point that was raised as well by reviewer 1; it was weak in the previous version and has not improved during the revision. I consider the experiment to be critical, in light of the exciting results the authors provide for different species. The generation of a new transgenic line is an enterprise that can take longer that the time allowed for a revision, but I understand that the authors already have the tools to go for it? ("… a new construct we have developed"). Furthermore, the mosaic approach followed by the authors in their initial submission allows the analysis in the injected generation and therefore within days, so I would have expected this point to be tackled experimentally. In their current form, the lines 320 to 330 are the most problematic of the entire manuscript, since they show only a weak trend (and not a "small increase" in proliferation, since the statistical analysis indicates that the differences are not significative). Presenting additional data in a subsequent "Research Advance" seems adequate for other experiments required by reviewer 1 and myself, but I feel this point is central to the story and I can not picture major problems in performing transient experiments to strengthen the proposed statement – keeping the proliferative state impairs RPE flattening. Is it realistic that the authors perform a transient over-expression of ccnd1 using an RPE specific promoter? – a construct they already have. If the authors are experiencing restrictions to the lab due to the pandemic and it is not possible to perform additional experiments, then I would suggest removing the lines 320 to 330. The manuscript would miss an interesting, important aspect but will gain in accuracy.I would strongly encourage the authors to tackle this point in the present version of the manuscript, though. If so, I would ask the authors to consider reporting this dataset at the end of the manuscript – after the report on the difference species in Figure 7. In this manner, they could show: (a) first, the inverse correlation between flattening and proliferation across vertebrates, (b) then prove that in one model species the switch can be induced experimentally.

We appreciate the reviewer’s concern and agree with her/him that the results we reported of forced cell proliferation in the zebrafish RPE were rather weak. We also felt that forcing proliferation in the zebrafish RPE would have allowed to “switch the zebrafish RPE” to “an amniote RPE”. We also could “not picture major problems in performing transient experiments to strengthen the proposed statement”, as the reviewer indicates. We proved ourselves wrong and perhaps for reasons that, eventually, should have been evident. To manipulate gene expression specifically in the forming RPE, we modified the original *E1-bhlhe40:GFP* construct contained in the ZED vector, replacing the GFP sequence for a multicloning site that allows for insertion of any gene of interest. To test that this modification did not interfere with tissue specificity, we re-inserted GFP and generated F0 embryos, in which the fluorescent signal was detected only in RPE cells. We then generated and injected an *E1-bhlhe40:ccnd1-HA* construct and determined the expression of cyclin D1 with antibodies against the HA tag. When the construct was injected in the embryos, we could detect only very few and sparse HA-positive cells in the RPE of the folding OV. These cells incorporated also EdU, although this was not always the case. More importantly, the large majority of HA+/EdU+ cells we detected were isolated and only very rarely we have observed small clusters of HA positive cells (2 or 4, see Author response image 1), strongly suggesting that despite cyclin D1 expression cells were unable to undergo a mitotic division. However, when they did, the RPE became “distorted” (Author response image 1). We have injected hundreds of embryos, analyzing them at slightly different time points, thinking that a specific time window was needed. However, the results were very similar across all time windows and clusters of 2/4 cells were rare. By injecting the construct in the already available Tg(E1-bhlhe40:GFP) line, we have observed that *ccnd1-HA* presence is associated with the lack of *bhlhe40* reporter expression (Author response image 1), suggesting that the absence of enhancer activation may preclude further cyclin D1 expression. To our surprise, we have not been able to find viable funders when we attempted to raise a stable transgenic Tg(*E1-bhlhe40:ccnd1-HA*) line, which needed several months. We have crossed already a large number of grown up fishes with no success.

Although we cannot exclude that a different experimental design may provide the expected results, we have concluded that expression of a single proliferative gene (even as strong as *ccdn1*) is insufficient to force cell proliferation in the RPE. The best explanation is likely because the RPE differentiates “en bloc”, explaining why, at difference of the amniote RPE, the zebrafish one has not been reported to acquire a neural retina fate, as we discuss in our manuscript. Further explanations may relate to the lack of the enhancer activity as we have explained above, or because forced proliferation induces rapid cell death preventing the expansion of cell proliferation. Some apoptotic cells were for example observed when we used the *UAS:ccd1* in the *Rx3Gal4* line, perhaps explaining, in retrospective, the weak results.

Whatever the reason might be, we cannot provide the evidence that we (and the reviewer) were seeking for. We have thus taken the reviewer’ suggestion of removing lines 320 to 330 from the manuscript (the description of forced expression of cyclin D1). Although frustrating, this seems the easiest option. To reinforce the message that proliferation has a limited contribution to RPE enlargement, we have emphasized an experiment performed by the lab of S. McFarlane (ref 22) in which they show that blocking cell proliferation has no effect of the enlargement of the RPE.

**Author response image 1. sa2fig1:** Effects observed after tissue specific overexpression of ccnd1 in the RPE. (**A**) Frontal cryostat section of 21 hpf Tg(*E1-bhlhe40:GFP*) embryo injected with the *E1-bhlhe40:ccnd1-HA* construct and immunestained for HA (magenta). (**B-C’’**) High power views of the regions boxed in A. Note the absence of reporter expression (green) in cells that express cyclinD1-HA (B, C). Note also the presence of isolated HA+ cells (C’, C’’) or that of a small HA+ clone that disrupts the flat morphology of the RPE (B’, B’’).